# Photodegradation of carbon dots cause cytotoxicity

Yue-Yue Liu[1,3], Nan-Yang Yu [1,3], Wen-Di Fang[1,3], Qiao-Guo Tan [2], Rong Ji [1], Liu-Yan Yang[1], Si Wei [1✉], Xiao-Wei Zhang [1✉] & Ai-Jun Miao [1✉]

Carbon dots (CDs) are photoluminescent nanomaterials with wide-ranging applications. Despite their photoactivity, it remains unknown whether CDs degrade under illumination and whether such photodegradation poses any cytotoxic effects. Here, we show laboratory-synthesized CDs irradiated with light degrade into molecules that are toxic to both normal (HEK-293) and cancerous (HeLa and HepG2) human cells. Eight days of irradiation photolyzes 28.6-59.8% of the CDs to <3 kilo Dalton molecules, 1431 of which are detected by high-throughput, non-target high-performance liquid chromatography-quadrupole time-of-flight mass spectrometry. Molecular network and community analysis further reveal 499 cytotoxicity-related molecules, 212 of which contain polyethylene glycol, glucose, or benzene-related structures. Photo-induced production of hydroxyl and alkyl radicals play important roles in CD degradation as affected by temperature, pH, light intensity and wavelength. Commercial CDs show similar photodegraded products and cytotoxicity profiles, demonstrating that photodegradation-induced cytotoxicity is likely common to CDs regardless of their chemical composition. Our results highlight the importance of light in cytocompatibility studies of CDs.

[1] State Key Laboratory of Pollution Control and Resource Reuse, School of the Environment, Nanjing University, Nanjing 210023 Jiangsu Province, China. [2] Key Laboratory of the Coastal and Wetland Ecosystems of Ministry of Education, College of the Environment and Ecology, Xiamen University, Xiamen 361102 Fujian Province, China. [3] These authors contributed equally: Yue-Yue Liu, Nan-Yang Yu, Wen-Di Fang. ✉email: weisi@nju.edu.cn; zhangxw@nju.edu.cn; miaoaj@nju.edu.cn

Carbon dots (CDs) were first described in 2004 as one of the impurities isolated from the purification of single-walled carbon nanotubes through preparative electrophoresis[1]. These "zero-dimensional" photoluminescent carbon nanomaterials are <10 nm in size and have promising applications in sensing, bioimaging, drug delivery, photodynamic therapy, and photo- and electro-catalysis[2,3]. A variety of methods to synthesize CDs have since been developed, including the "bottom-up" dehydration of organic compounds (e.g., carbohydrate) and the "top-down" cutting of graphite, carbon fiber, carbon nanotubes, and other carbon sources[4,5]. In contrast to traditional semiconductor quantum dots, which contain toxic heavy metals (such as Cd and Pb), and organic dyes, CDs have high aqueous solubility, are biocompatible, chemically inert, and can be easily functionalized[6,7]. Furthermore, they have tunable emissions and large two-photon excitation cross-section, making them more attractive than traditional quantum dots[3,5].

The cytotoxicity of CDs have been extensively examined using different mammalian cell lines[6,8], and different CD composition and surface functionalization[9]. They have been deemed intrinsically non-cytotoxic and therefore, suitable for various biomedical applications[9]. The problem, however, is that most of the cytotoxicity studies up until now have been done in the dark[10]. Many of the proposed applications of CDs involve light (e.g., bioimaging, photocatalysis, light-emitting diodes, photovoltaic cells etc.)[11,12]. Further, CDs are photoactive; recent studies have shown that light-irradiated CDs produce reactive oxygen species[13,14] that are toxic to bacteria[15,16] and yeasts[17]. Despite their photoactivity, it remains unknown whether CDs degrade under illumination and whether such photodegradation poses any cytotoxic effects on human cells.

Here, we show that laboratory-synthesized CDs irradiated with light degrade into molecules that are toxic to both normal (HEK-293 kidney epithelial cells) and malignant (HeLa cervical cancer cells and HepG2 hepatocellular carcinoma cells) human cells. CDs are synthesized using a common glucose pyrolysis method and irradiated with light under different conditions (e.g., different pH, ionic strength, temperature, light intensity, and wavelength) for different durations. Potential role of free radicals in CD photodegradation is revealed. Through reduced human transcriptome (RHT) modeling of 1200 human genes that responded to the laboratory-synthesized CDs in a concentration-dependent manner, we identify the molecular pathways that are perturbed by the CDs. Comparing the genes and pathways from the RHT profiles with the Comparative Toxicogenomics Database (CTD) further reveals that the key toxicants likely have a high degree of unsaturation and contain benzene, carbonyl, or hydroxyl groups. We analyze the photolyzed molecules using high-throughput, nontarget high-performance liquid chromatography–quadrupole time-of-flight mass spectrometry (HPLC-QTOF) and visualize their correlation with cytotoxicity through molecular network and community analysis. We detect 1431 molecules, 499 of which are related to cytotoxicity. Using an automatic homologue and high-frequency fragment analysis, the structures of 212 cytotoxicity-related photodegradation products are identified. Commercial CDs prepared using different methods also produce similar photodegradation products and cytotoxicity profiles, suggesting that these are likely general characteristics of CDs. Contrary to previous claims of biocompatibility, our results show CDs photodegrade and release substances that can induce cytotoxicity in human cells.

## Results and discussion
### Physicochemical characterization of lab-synthesized CDs.
Transmission electron microscopy (TEM) images (Supplementary Fig. 1a) showed that our laboratory-synthesized CDs had an average size of 3.0 nm, comparable to that determined by Zhu et al.[18] (2.75–3.65 nm) for CDs synthesized using a similar method. Lattice spacing of 0.21 nm as measured from high-resolution TEM images (Supplementary Fig. 1a inset) matched the in-plane lattice spacing of graphite (100 facet)[19,20], indicating the core of our CDs is graphitic. The size of CDs was further verified by atomic force microscopy (AFM) with an average size of 2.5 nm (Supplementary Fig. 1b). The two absorption peaks at 224 and 280 nm in the UV–Vis spectrum (Supplementary Fig. 1c) originate from $\pi$–$\pi^*$ transitions of the C–C and C=C bonds in the aromatic $sp^2$ domains of the core and n–$\pi^*$ transition of multiconjugate C=O and C–O on the CD surface, respectively[21,22]. Their excitation–emission spectra (Supplementary Fig. 1c) were characterized by a red shift (from 469 to 516 nm) in the emission wavelength when the excitation wavelength increased from 380 to 440 nm. Excitation-dependent photoluminescence may result from changes in the relative intensity of a few emission species, optical selection of differently sized nanoparticles, and/or the different emissive traps on the surface of the CDs[4,23].

The peak at 1385.6 cm$^{-1}$ (D band) in the Raman spectra of the CDs shown in Supplementary Fig. 1d originated from the vibrations of carbon atoms with dangling bonds in the termination plane of disordered graphite or glassy carbon[24]. The G band at 1581.4 cm$^{-1}$ corresponded to the $E_{2g}$ mode of graphite and was related to the vibration of $sp^2$-bonded carbon atoms in a two-dimensional hexagonal lattice[2]. The intensity ratio of the D and G bands ($I_D/I_G$), a measure of the extent of disorder, was calculated to be 0.4, indicative of increased structural defects compared to graphite[25]. The broad peak at 21.4° [corresponding to the (002) peak of graphite] in the X-ray diffraction (XRD) spectra further confirmed the disordered structure of the CDs (Supplementary Fig. 1e). In this case, the calculated interlayer spacing of the CDs (0.41 nm) was higher than that of graphite (0.34 nm), indicating poor crystallization of our CDs[2,26]. Fourier transform infrared spectroscopy (FTIR) studies (Supplementary Fig. 1f) showed stretching vibrations of O–H (3370 cm$^{-1}$), C–H (2915 and 2875 cm$^{-1}$), and C=O (1702 cm$^{-1}$), asymmetric stretching vibrations of C–O–C (1060 cm$^{-1}$), and bending vibrations of =C–H (935 and 902 cm$^{-1}$)[18,27]. In the X-ray photoelectron (XPS) spectrum (Supplementary Fig. 1g), only C and O peaks were present; elemental analysis further showed that our CDs contained C (63.6%), O (31.1%), and H (5.3%). Deconvoluting the C1s and O1s peaks (Supplementary Fig. 1h, i) revealed the presence of C–C, C=C, –CH$_2$–O, C–O, C=O, O=C–O, and O–H[27,28]. The abundant hydrophilic functional groups on the CD surface allowed them to disperse well in the experimental medium. Their average hydrodynamic diameter was 11.7 nm as obtained from a dynamic light scattering particle sizer. Together, these results confirmed that our laboratory-synthesized CDs are composed of a graphitic structure with poor crystallization, which is attributed to the generation of oxygen containing groups.

### Photodegradation of the CDs.
We investigated the effects of various environmental factors (i.e., light intensity and wavelength, temperature, pH, and ionic strength) and particle size on the photodegradation kinetics of the laboratory-synthesized CDs. As shown in Fig. 1a, CD degradation increased with light intensity. Negligible degradation occurred in the dark while 19.1%, 25.7%, 34.0%, and 42.8% of the CDs degraded into dissolved molecules <3 kilo Daltons (kD) in size after 8 days of irradiation to white fluorescent light with the intensity of 15, 30, 60, and 90 µmol photons/m$^2$/s, respectively. As for wavelength, CD degradation increased as the light wavelength decreased from 620 nm (red) to

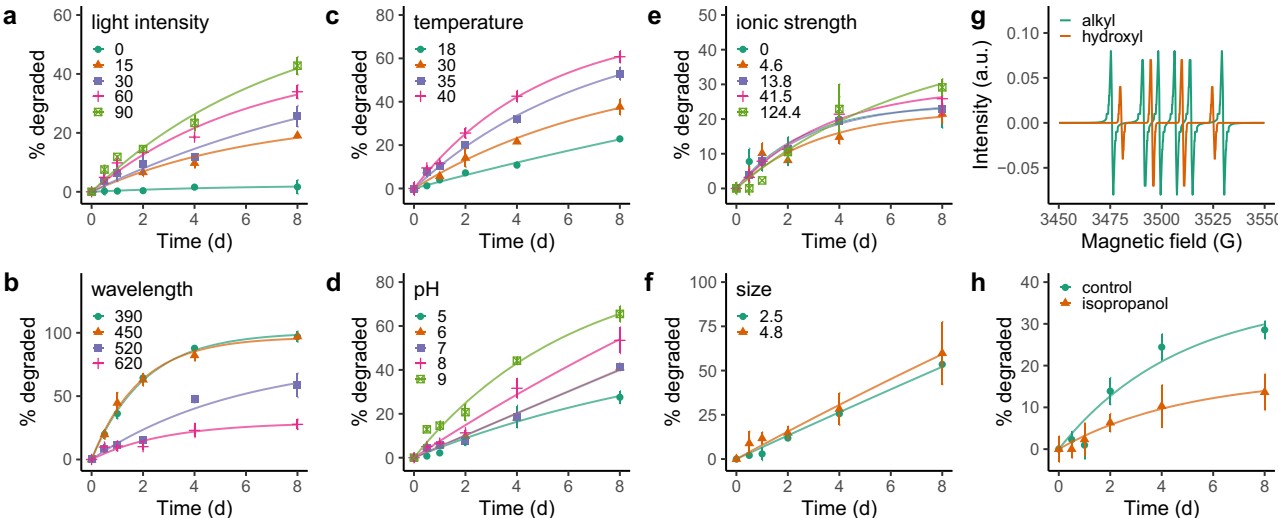

**Fig. 1 Photodegradation of laboratory-synthesized carbon dots (CDs). a–d** More CDs degraded at higher light intensity (0, 15, 30, 60, and 90 μmol photons/m$^2$/s) (**a**), shorter light wavelength (390, 450, 520, and 620 nm) (**b**), higher temperature (18, 30, 35, and 40 °C) (**c**), and pH (5, 6, 7, 8, and 9) (**d**). **e** Ionic strength (0, 4.6, 13.8, 41.5, and 124.4 mM) had no effects on CD degradation. **f** CDs with different sizes (2.5 and 4.8 nm) show similar degradation kinetics. **g** Free hydroxyl and alkyl radicals were produced when CDs were irradiated by white fluorescent light (60 μmol photons/m$^2$/s). **h** When the hydroxyl radicals were scavenged by 10 mM isopropanol, less CDs degraded as compared to the control treatment without any addition of isopropanol. Data are presented as the mean ± s.d. ($n = 3$ independent experiments). Source data are provided as a Source Data file.

450 nm (blue) and remained constant thereafter (Fig. 1b). In this case, nearly all CDs (97.1–96.9%) degraded after 8 days of irradiation to blue (450 nm) and purple (390 nm) light. Additionally, more CDs degraded at higher temperature (18–40 °C, Fig. 1c) or pH (5–9, Fig. 1d). By contrast, ionic strength (0–124.4 mM, Fig. 1e) and particle size (2.5 and 4.8 nm, Fig. 1f) had no significant (two-way ANOVA, $p = 0.51$ and 0.87 for ionic strength and particle size, respectively) effects on CD degradation.

To examine the mechanisms underlying CD photodegradation, we used electron paramagnetic resonance spectroscopy (EPR) in combination with two spin traps 5,5-dimethyl-1- pyrroline-N-oxide (DMPO) and 2,2,6,6-tetramethylpiperidine (TEMP) to detect the production of reactive free radical species during degradation. As shown in Fig. 1g, both hydroxyl and alkyl radicals but no singlet oxygen were formed when the CDs were irradiated with white fluorescent light for 10 min. Contrastively, no radicals were detected in the dark or in the medium with the addition of <3 kD molecules degraded from the CDs only. These phenomena imply that the free radical species were produced by the CDs themselves under irradiation. To further explore how these radicals might be involved in CD photodegradation, we examined the degradation kinetics of the CDs in the presence of a hydroxyl radical scavenger (i.e., isopropanol). The degradation of the CDs in the dark with and without the presence of hydroxyl radicals as produced by the Fenton reaction was also compared. As shown in Fig. 1h, CD degradation decreased significantly (two-way ANOVA, $p = 0.001$) in the presence of isopropanol. Further, more CDs degraded when FeCl$_2$ (0.3–3 mM) was added with its molar ratio to H$_2$O$_2$ of 1:1 and the proportion of CD degradation increased with the increased production of hydroxyl radicals (Supplementary Fig. 2a, b). These results suggest the importance of hydroxyl radicals in CD photodegradation. Similar experiment cannot be performed for alkyl radicals as no effective scavenger is available due to their complicated speciation and their production cannot be simply manipulated as that for hydroxyl radicals. Nevertheless, since isopropanol cannot completely prevent CD photodegradation, alkyl or other radicals may also play important roles. Overall, upon light irradiation, photogeneration of electrons and holes would occur on the surface of the CDs[13], resulting in

the formation of hydroxyl and alkyl radicals (Supplementary Fig. 3). Conjugated π-bond, surface defects, and functional groups of the CDs were also possibly involved in the generation of radicals[29]. The radicals could then attack the CDs in multiple ways (e.g., transferring electrons to CDs, abstracting H-atom from CDs, hydroxylating CDs by attacking the C=C bonds of CDs)[30,31] and the CDs thus degraded.

**Cytotoxicity of the CDs.** To investigate the cytotoxicity of our CDs, we exposed three different cells lines (HeLa, HepG2, and HEK-293) to different concentrations (0, 10, 30, 100, and 300 mg carbon/L) of CDs that have been irradiated with white fluorescent light (60 μmol photons/m$^2$/s) for 0.5 (i$_{0.5}$-CD), 1 (i$_1$-CD), 4 (i$_4$-CD), and 8 (i$_8$-CD) days, and CDs that have not been irradiated (n-CD). No inhibition of cell viability was observed in cells exposed to n-CD (Fig. 2a, b); in HEK-293 cells, cell viability was even improved with increasing CD concentrations (Fig. 2c). These results showed that the CDs themselves were biocompatible, consistent with the literature[3,5]. However, with irradiated CDs, cytotoxicity increased with longer irradiation time (Fig. 2a–c). The viability of HeLa, HepG2, and HEK-293 cells reduced by 91.0%, 49.0%, and 90.0%, respectively, when cells were exposed to the highest concentration (300 mg carbon/L) of i$_8$-CDs. The light-induced cytotoxicity of CDs is likely due to either the production of reactive oxygen species[13,14] as reported in bacteria and yeasts, or the release of cytotoxic photolyzed products during irradiation or both.

To understand how CD photodegradation might be involved in their light-induced cytotoxicity, we then ultrafiltered the i$_8$-CD through a 3 kD regenerated cellulose membrane and compared the cytotoxicity of both the <3 kD fraction containing photolyzed molecules (i$_8$-CD$_{<3\,kD}$) and the >3 kD fraction containing degraded CDs (i$_8$-CD$_{>3\,kD}$) with that of the total unfiltered suspension of i$_8$-CD. As shown in Fig. 2d–f, i$_8$-CD$_{>3\,kD}$ exhibited negligible cytotoxicity, especially to HeLa and HepG2 cells. In contrast, the cytotoxicity of i$_8$-CD$_{<3\,kD}$ was comparable to (in the case of HepG2 cells) or higher than (in HeLa and HEK-293 cells) that seen with the total unfiltered i$_8$-CD suspension. These results

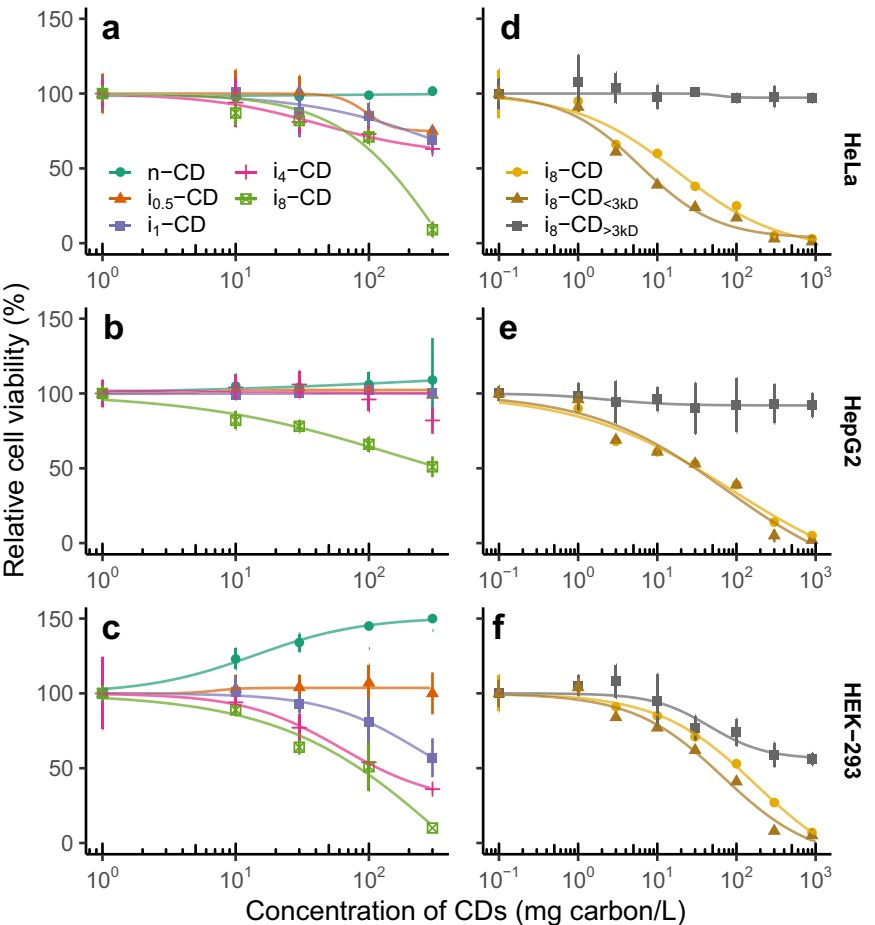

**Fig. 2 Cell viability testing of laboratory-synthesized carbon dots (CDs). a–c** Dose–response data show the cytotoxicity of CDs to HeLa (**a**), HepG2 (**b**), and HEK-293 (**c**) cells increased with irradiation time. The three cell lines were exposed to different concentrations (0, 10, 30, 100, and 300 mg carbon/L) of CDs that have been irradiated with white fluorescent light (60 μmol photons/m²/s) for 0 (n-CD), 0.5 (i$_{0.5}$-CD), 1 (i$_1$-CD), 4 (i$_4$-CD), and 8 (i$_8$-CD) days. **d–f** Dose–response data show the photolyzed products in the <3 kD fraction contributed substantially to the photo-induced cytotoxicity of CDs to HeLa (**d**), HepG2 (**e**), and HEK-293 (**f**) cells. The cytotoxicity of both the <3 kD fraction containing photolyzed molecules (i$_8$-CD$_{<3\,kD}$) and the >3 kD fraction containing degraded CDs (i$_8$-CD$_{>3\,kD}$) was compared with that of the total unfiltered suspension of i$_8$-CD at concentrations of 0, 1, 3, 10, 30, 100, 300, and 900 mg carbon/L. Data are presented as the mean ± s.d. ($n = 6$ independent experiments). Source data are provided as a Source Data file.

demonstrate that the photolyzed products in the <3 kD fraction contributed substantially to the photo-induced cytotoxicity of the CDs.

We further studied the toxicity of the photodegraded CDs in a concentration-dependent RHT experiment[32]. Briefly, we exposed HepG2 cells to seven fivefold dilutions of n-CD (0.019–300 mg carbon/L), i$_8$-CD (0.00013–2 mg carbon/L), i$_8$-CD$_{<3\,kD}$ (0.00019–3 mg carbon/L), and i$_8$-CD$_{>3\,kD}$ (0.019–300 mg carbon/L) in a single replicate. Four vehicle controls (0.1% v/v of methanol) were included. After 24 h of CD exposure, we isolated the total RNA from the cells and subjected them to high-throughput sequencing and obtained a dose–response curve for each gene (see "Methods" for details).

Sequence counts ranging from 543,084 to 5,848,694 were obtained (Supplementary Fig. 4). All sequence depths were >300,000 reads, a level sufficient to detect the signals (counts of >5) of at least 750 genes, as calculated by Monte Carlo simulations[32]. The number of differentially expressed genes (DEGs) identified in nine dose–response models (Supplementary Table 1) ranged from 65 to 172 across the four samples (see "DEGs-n-CD", "DEGs-i8-CD", "DEGs-i8-CD > 3 kD", and "DEGs-i8-CD < 3 kD" worksheets in Supplementary Excel File). Based on the effective concentration for each DEG (EC$_{DEG}$) determined from the dose–response curves, we found i$_8$-CD and

i$_8$-CD$_{<3\,kD}$ to be more toxic than n-CD and i$_8$-CD$_{>3\,kD}$, which is consistent with the cell viability results in Fig. 2. As shown in Fig. 3a, the EC$_{DEG}$ values were 0.00017–3.9 mg carbon/L for i$_8$-CD, 0.00019–28.4 mg carbon/L for i$_8$-CD$_{<3\,kD}$, 0.092–1066.7 mg carbon/L for n-CD, and 0.087–1150 mg carbon/L for i$_8$-CD$_{>3\,kD}$. A comparison of the DEGs of the two most toxic samples (i$_8$-CD and i$_8$-CD$_{<3\,kD}$) revealed 15 DEGs in common, most of which were cancer-related (red circle in Fig. 3d). These DEGs included *ETS1*[33], *POU2F1*[34], *IGF1R*[35], *SDC1*[36], and *PDCD6*[37], which have been implicated in breast cancer, and *PPP2CA*[38] and *FGFR3*[39], which are relevant to lung cancer. *SERP1*[40] is relevant to pancreatic cancer and *SMUG1*[41] to colorectal cancer. Moreover, *BCL6*[42] as an oncogene is found in B-cell lymphomas, and *FDPS*[43] is highly expressed in tumors by canonical oncogenic signaling pathways. Other DEGs in this group were *IL8*[44], which mediates the neutrophil signaling, *DMD*[45], which instructs the synthesis of dystrophin, and *FOXO1*[46] and *NR1H2*[47], which play important roles in the regulation of cholesterol, fatty acid, and glucose homeostasis.

When the RHT profiles were subjected to a Gene Ontology pathways (GO)-based analysis to identify the biological pathways involved in CD-induced cytotoxicity, 398 enriched GO terms were obtained (see "GO-n-CD", "GO-i8-CD", "GO-i8-CD > 3kD", and "GO-i8-CD < 3kD" worksheets in Supplementary

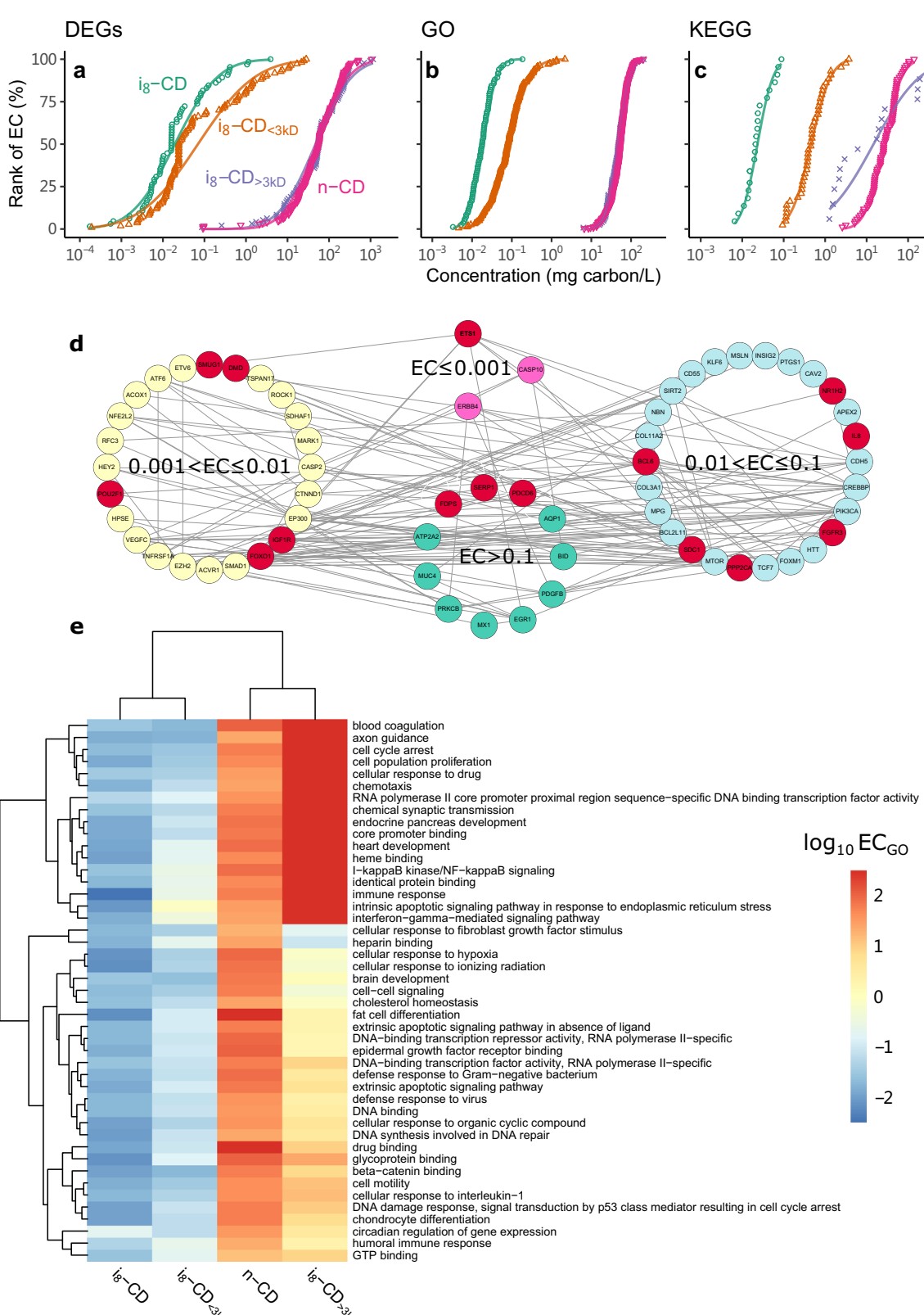

**Fig. 3 Cytotoxicity of laboratory-synthesized carbon dots (CDs) as obtained from reduced human transcriptome (RHT). a–c** Cumulative distribution of the effective concentration (EC) for differentially expressed genes ($EC_{DEG}$) (**a**), Gene Ontology ($EC_{GO}$) (**b**), and Kyoto Encyclopedia of Genes and Genomes ($EC_{KEGG}$) (**c**) pathways confirm the cytotoxicity of the total unfiltered suspension of 8-day irradiated CDs ($i_8$-CD) and its <3 kD fraction ($i_8$-$CD_{<3\,kD}$) was much higher than that of its >3 kD fraction ($i_8$-$CD_{>3\,kD}$) and the nonirradiated CDs (n-CD). Please refer to Methods for detailed definition of $EC_{DEG}$, $EC_{GO}$, and $EC_{KEGG}$. **d** Concentration-dependent gene network of DEGs identified by RHT in HepG2 cells exposed to $i_8$-CD. The DEGs were grouped according to their effective concentration (EC, mg carbon/L). The 15 DEGs co-detected in HepG2 cells exposed to $i_8$-CD and $i_8$-$CD_{<3\,kD}$ were highlighted in red circles. **e** Hierarchical clustering of the 45 co-enriched GO terms reveal similar modes of action for $i_8$-CD and $i_8$-$CD_{<3\,kD}$. Source data are provided as a Source Data file.

Excel File). The effective concentration of each GO term ($EC_{GO}$) was then calculated as the geometric mean of the $EC_{DEG}$ of the matched DEGs. Similar to the $EC_{DEG}$ results, the $EC_{GO}$ values of $i_8$-CD and $i_8$-$CD_{<3\,kD}$ were approximately three orders of magnitude lower than those of n-CD and $i_8$-$CD_{>3\,kD}$ (Fig. 3b), suggesting that the relative toxic potency of $i_8$-CD and $i_8$-$CD_{<3\,kD}$ was much higher than n-CD and $i_8$-$CD_{>3\,kD}$. Among the 398 GO terms, 45 biological process and molecular function terms, including inflammation, endocrine disruption, xenobiotics metabolism, DNA damage, and immune response pathways, were co-enriched in HepG2 cells exposed to $i_8$-CD and $i_8$-$CD_{<3\,kD}$. Further hierarchical clustering of the 45 co-enriched GO terms revealed similar modes of action for $i_8$-CD and $i_8$-$CD_{<3\,kD}$ (Fig. 3e), demonstrating the role of $i_8$-$CD_{<3\,kD}$ in the observed photo-induced cytotoxicity.

The effective concentration of the responsive Kyoto Encyclopedia of Genes and Genomes (KEGG) pathways (see "KEGG-n-CD", "KEGG-i8-CD", "KEGG-i8-CD > 3 kD", and "KEGG-i8-CD < 3 kD" worksheets in Supplementary Excel File) ($EC_{KEGG}$) also demonstrated $i_8$-CD and $i_8$-$CD_{<3\,kD}$ were more cytotoxic than n-CD and i-$CD_{>3\,kD}$ (Fig. 3c). The 20 co-enriched KEGG pathways (red colored in "KEGG-i8-CD" and "KEGG-i8-CD < 3kD" worksheets of Supplementary Excel File) for $i_8$-CDs and $i_8$-$CDs_{<3\,kD}$ included TGF-β, ErbB, and Wnt, which are early molecular signals essential to cancer progression. The TGF-β signaling pathway[48] known to mediate tumor suppression is also a promoter of tumor progression and invasion. ErbB receptors[49] are overexpressed or mutated in many cancers, especially breast cancer, ovarian cancer, and non-small cell lung cancer. Wnt signaling[50] has been closely linked to colorectal cancer. Although KEGG pathways such as Wnt and ErbB were also associated with n-CD and $i_8$-$CD_{>3\,kD}$, their $EC_{KEGG}$ values were much higher than those of $i_8$-CD and $i_8$-$CD_{<3\,kD}$, indicating that a larger number of toxicants was present in the <3 kD fraction.

Potential carcinogenic effects of the CD photodegradation products were further evidenced by the results of cell invasion and transformation (as indicators of carcinogenicity) assays (see Supplementary Information for details). Both $i_8$-CD and $i_8$-$CD_{<3\,kD}$ showed remarkable inductive effects on the invasion and transformation of HepG2 cells especially in the two lower concentration treatments while no obvious effects were observed for n-CD and $i_8$-$CD_{>3\,kD}$ (Supplementary Fig. 5). The highest concentrations of $i_8$-CD and $i_8$-$CD_{<3\,kD}$ used herein were their 20% effective concentration for the inhibition of cell viability. The significant cytotoxicity of the highest concentration treatments of $i_8$-CD and $i_8$-$CD_{<3\,kD}$ explains their much less inductive effects on cell invasion and transformation as compared to the other two lower concentration treatments. Nevertheless, further systematic studies were needed to verify the possible carcinogenic effects of CD photodegradation products.

Besides identifying cytotoxic pathways, RHT profiles also provide structural information on the toxicants based on the associations between chemicals and genes/pathways in the CTD (i.e., the Comparative Toxicogenomics Database). In our study, among the 223 chemicals obtained from the dose-dependent DEGs (see "DEG-based chemical list" worksheet in Supplementary

Excel File), ten organic compounds (red colored in the above worksheet) were identified as capable of affecting >50% of the DEGs in cells exposed to $i_8$-CD and $i_8$-$CD_{<3\,kD}$. Similarly, among the 3713 chemicals identified from the 45 co-enriched GO terms or 20 KEGG pathways in HepG2 cells incubated with $i_8$-CD and $i_8$-$CD_{<3\,kD}$ (see "pathway-based chemical list" worksheet in Supplementary Excel File), 486 organic compounds (red colored in the above worksheet) were potentially capable of affecting >50% of the GO and KEGG pathways. These 486 chemicals included all ten chemicals derived based on the DEGs. As shown in Supplementary Table 2, benzene, carbonyl, and hydroxyl groups were the most frequently detected structures among these ten molecules, suggesting that these functional groups might be present in the main toxic substances of $i_8$-CD and $i_8$-$CD_{<3\,kD}$.

**Identification of the degradation products.** Given the <3 kD fraction had a main contribution to the cytotoxicity of the irradiated CDs, we analysed the molecular structures of the photo-lyzed molecules (<3 kD fraction) of all irradiated CD samples (i.e., $i_{0.5}$-CD, $i_1$-CD, $i_4$-CD, and $i_8$-CD) using a high-throughput, nontarget method. n-CD was also included as control. The intensity of the total ion chromatograph (Supplementary Fig. 6) as well as the peak area and peak number (Fig. 4a) increased when CDs were irradiated longer; a total of 1886 peaks corresponding to 1431 photodegradation products were detected in $i_8$-CD. Moreover, the intensity/area of the peaks showed different patterns of variation with irradiation time (Fig. 4b) and the number of peaks with an $m/z$ between 270 and 350 was larger than other peaks (Fig. 4c). As the peak area and peak number are indicators of the concentration and number of photodegradation products, these results suggest that the irradiated CDs released many small molecular substances (Fig. 4c) that had different formation kinetics (Fig. 4b).

The formation kinetics of photodegradation products indicate how quickly degradation reactions occur and this information can be used to cluster products that are generated via similar degradation pathways[51,52]. Thus, to visualize the similarities in the formation kinetics of the different degradation products, we conducted a network analysis (see "Methods" for details). Of the 1431 photodegradation products (represented as nodes) as shown in Fig. 4d, 732 of them were correlated significantly ($p < 0.001$, see Source Data file for detailed $p$ values) to each other in their formation kinetics, and 3964 edges (represented as links) between the different products were established. Accordingly, 12 communities [nine of them with significant ($p < 0.05$, see Source Data file for detailed $p$ values) correlation to cytotoxicity were numbered in Fig. 4d as described below] indicative of the connectivity between different nodes were obtained in which >10 nodes with similar formation kinetics clustered together[53]. To further identify the photodegradation products (nodes) associated with cytotoxicity, a correlation analysis was performed between the time-related variation of cytotoxicity and the formation kinetics of each product (node) in Fig. 4d. The 1431 nodes shown in Fig. 4d are colored according to their correlation with the

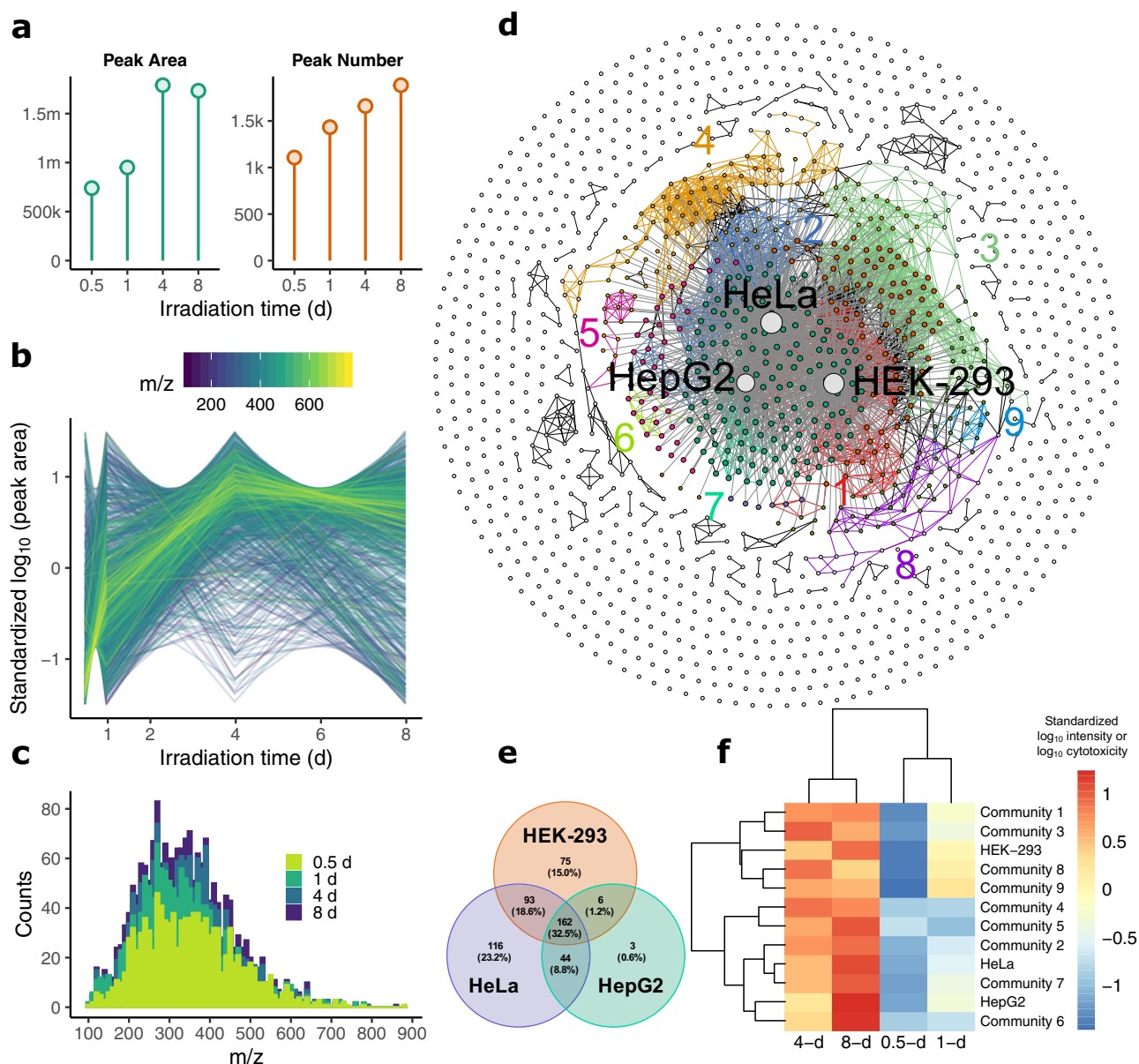

**Fig. 4 Identification of the degradation products of laboratory-synthesized carbon dots (CDs). a** The peak area and peak number of the photodegradation products increased with irradiation time (0.5, 1, 4, and 8 days). **b** The area/intensity of the peaks showed different patterns of variation with irradiation time. **c** The number of the peaks with an *m/z* between 270 and 350 was larger than other peaks. **d** Network analysis of the similarities in the formation kinetics of the 1431 photodegradation products/nodes obtained from a quadropole time-of-flight mass spectroscopy as well as the correlation between the formation kinetics of the products and time-related change in cytotoxicity. Twelve communities [nine of them with significant ($p < 0.05$, two-tailed Pearson correlation analysis, see Source Data file for detailed $p$ values) correlation to cytotoxicity were numbered] indicative of the connectivity between different nodes were obtained in which >10 nodes with similar formation kinetics clustered together. The products/nodes are differently colored (gray circle—uncorrelated to cytotoxicity, dark green circle—correlated to the cytotoxicity in all three cells, the other colored circles—correlated to the cytotoxicity in one or two cells) according to their correlation with the cytotoxicity in HEK-293, HeLa, and HepG2 cells. Based on the correlation intensity of each peak with cytotoxicity, three cell nodes were added, clarifying the significant ($p < 0.05$, two-tailed Pearson correlation analysis, see Source Data file for detailed $p$ values) association between the products in 9 of the 12 communities and the cytotoxicity in at least one cell line. Products/nodes in the nine communities were linked by lines with their color the same as that of the community number. **e** Distribution of the 499 cytotoxicity-correlated products in the Venn diagram of HepG2, HeLa, and HEK-293 cells. 162 products were correlated with the cytotoxicity in all three cell lines. **f** Clustering analysis of the correlation between the cytotoxicity in the three cell lines and the nine communities numbered in **d**. Source data are provided as a Source Data file.

cytotoxicity in HEK-293, HeLa, and HepG2 cells (see Fig. 4 for details). According to Fig. 4d, 499 of the 1431 photodegradation products were associated with cytotoxicity in at least one cell line: 215 products in HepG2 cells, 415 in HeLa cells, 336 in HEK-293 cells, and 162 products that correlated with cytotoxicity in all three cell lines (Fig. 4e). Based on the correlation intensity of each

product with cytotoxicity, three cell nodes were added into Fig. 4d, clarifying the significant ($p < 0.05$, see Source Data file for detailed $p$ values) association between the products in 9 of the 12 communities and cytotoxicity in at least one cell line. We further calculated the correlation between the time-dependent cytotoxicity reflected by these three cell-line-based nodes and the

**Table 1 Structures or substructures of the identified photodegradation products from laboratory-synthesized carbon dots.**

| Group | n | Structure or substructure | Group | n or n1+n2 | Structure or substructure |
|---|---|---|---|---|---|
| 1 | 3-11 | $HO-[C_2H_4O]_n-H$ | 12 | 6-12 | $HO-CHO-[C_2H_4O]_{n1}-H$ / $CHO-[C_2H_4O]_{n2}-H$ / $CH_2OH$ |
| 2 | 4-14 | $Glc-[C_2H_4O]_n-H$ | 13 | 4-13 | $HOH_2C-\overset{HOH}{C}-CH_2O-[C_2H_4O]_n-H$ |
| 3 | 4-7 | $(Glc-H_2O)-[C_2H_4O]_n-H$ | 14 | 3-8 | $OHC-\overset{HOH}{C}-CH_2O-[C_2H_4O]_n-H$ |
| 4 | 2-4 | $(Glc-2OH)-[C_2H_4O]_n-H$ | 15 | 5-16 | $CH_2O-[C_2H_4O]_{n1}-H$ / $CHO-[C_2H_4O]_{n2}-H$ / $OH$ |
| 5 | 4-8, 10 | $(Glc-2H_2O-2H)-[C_2H_4O]_n-H$ | 16 | 5-10 | $OHC-\overset{HOH}{C}-O-[C_2H_4O]_n-H$ |
| 6 | 7,9 | $(Glc-3H_2O)-[C_2H_4O]_n-H$ | 17 | 8-10 | $C_2H_5O-[C_2H_4O]_n-H$ |
| 7 | 4-7, 9-11 | $(Glc-CH_2O)-[C_2H_4O]_n-H$ | 18 | 3-6 | $CH_3O-[C_2H_4O]_n-H$ |
| 8 | 5-7, 9 | $(Glc-CH_2O-O)-[C_2H_4O]_n-H$ | 19 | 4-6 | $C_7H_9O_3-[C_2H_4O]_n-H$ |
| 9 | 2-9 | $(Glc-CH_2O-2O)-[C_2H_4O]_n-H$ | 20 | 3-7 | $C_7H_7O_3-[C_2H_4O]_n-H$ |
| 10 | 3-9 | $HOC_2H_4O-\underset{H}{C}=CHO-[C_2H_4O]_n-H$ | 21 | 4-6 | $C_8H_9O_3-[C_2H_4O]_n-H$ |
| 11 | 3, 5-7 | $C_4H_7O_2-[C_2H_4O]_n-H$ | 22 | [a]N.A. | $R-[C_2H_4O]_n-H$ |

| Group | Structure or substructure |
|---|---|
| 23 | (quinolin-3-amine, $NH_2$), (6-hydroxyquinoline, $HO$-), (indol-3-yl-acetic acid, $OH$), [b]($C_6H_5-CH_2^+$), [b](phenyl cation) |

[a]N.A. means not available.
[b]All 16 substances in Group 23 contained one of these two substructures.

formation kinetics of the 9 communities. The correlations of the different communities with the cell lines differed, with Communities 6, 7, and 8/9 having the closest relationship with HepG2, HeLa, and HEK-293 cells, respectively (Fig. 4f). Following the molecular network analysis in Fig. 4d–f, the 499 photodegradation products (nodes) associated with cytotoxicity were subjected to a structural analysis.

The structures of 212 photodegradation products associated with cytotoxicity in at least one cell line were identified (see "Methods" for details), 65 products of which were associated with cytotoxicity in all cell types. Based on the homologue and high-frequency fragment analysis, the 212 identified products were divided into 23 groups according to their chemical structures (Table 1, also see "products identified in lab-CDs" worksheet in Supplementary Excel File). The association/link of the products

(represented as dots around the circle) in each group with cytotoxicity in each cell type (represented as dots inside the circle) is shown in Supplementary Fig. 7. The dot's size indicates the number of links.

From Table 1 and "products identified in lab-CDs" worksheet in Supplementary Excel File, we note that polyethylene glycol (PEG) with molecular weights between 150 and 502 is an important component in the structure of all chemicals identified in Groups 1–22. PEG is a surfactant and passivating agent added to improve the photoluminescence of CDs[18]. Because PEG chains were mostly located on the surface of the CDs, they underwent extensive photodegradation. The chemicals in Group 1 were solely PEG molecules ($HO–[C_2H_4O]_n–H$), with an $n$-value ranging between 3 and 11. Besides PEG, glucose and its derivatives were also important structural components for the chemicals in Groups

2–18. In this case, the chemicals in Group 2 were identified as PEG-conjugated glucose while fragments from the dehydration, dehydroxylation, or dehydrogenation of glucose were found in the MS/MS spectra of the chemicals in Groups 3–6. Glucose can be photodegraded to arabinose ($C_5H_{10}O_5$), erythrose ($C_4H_8O_4$), glyceraldehyde ($C_3H_6O_3$), glycol-aldehyde ($C_2H_4O_2$), glyoxal ($C_2H_2O_2$), or formaldehyde ($CH_2O$) as a result of the excitation of the antibonding orbitals of non-bonding electrons on the lactol oxygen atom[54,55]. Since PEG-conjugated glucose (Group 2) contains a lactol oxygen atom, $C_5H_{10}O_5$–PEG (Group 7) and $C_3H_6O_3$–PEG (Group 14) were formed during the photodegradation of CDs. Due to the presence of an aldehyde group in the glucose degradation products, $C_3H_6O_3$–PEG (Group 14) and $C_2H_4O_2$–PEG (not identified herein) were able to further react with PEG to form $C_3H_8O_4$–PEG$_2$ (Group 12) and $C_2H_6O_3$–PEG$_2$ (Group 15). $C_2H_4O_3$–PEG (Group 16) was the reaction product of glyoxal and PEG. Groups 8–11 included the deoxygenated products of pentose and tetrose conjugated with PEG. Chemicals in Groups 17–18 ($C_2H_6O$–PEG and $CH_4O$–PEG) could have been formed by the degradation of compounds in Groups 8–11. Group 13 comprised of compounds containing PEG-conjugated glycerol, a typical product of glucose catabolism. Compounds in Groups 19–21 had a higher degree of unsaturation than most chemicals in Groups 1–18 based on their formulas. Nevertheless, it was difficult to deduce their chemical substructures from their formulas because of their unidentified fragments. Group 22 contained 27 substances with fragments other than PEG that remained undetected; thus, only a formula with an unknown group R was assigned to the compounds in this group.

All MS/MS spectra of Group 23 included $[C_7H_7]^+$ or $[C_6H_5]^+$ (Table 1, also see "products identified in lab-CDs" worksheet in Supplementary Excel File), suggesting the presence of a benzene ring in the structures of all 16 chemicals in this group. Four chemicals in Group 23 were identified as $C_{12}H_{14}$, $C_{13}H_{14}$, $C_{14}H_{14}$, and $C_{24}H_{24}$. Although their structures were not confirmed, their molecular formulas, high degree of unsaturation, and ion fragments suggested that all four compounds were polycyclic aromatic hydrocarbons. Three other chemicals were identified as 3-aminoquinoline, 6-hydroxyquinoline, and indole-3-acetic acid based on comparisons of the MS/MS spectra with a public MS database. The identification results were consistent with the RHT data (Supplementary Table 2), where compounds responsible for half of the variation in the DEGs or GO/KEGG pathways contained benzene, carbonyl, or hydroxyl groups in their structures.

To unfold the roles of the different molecules from the 23 groups above in the photo-induced cytotoxicity of the CDs, we concentrated the degradation products by solid-phase extraction, divided the extract into nine fractions based on their retention time in HPLC (see Supplementary Information for details), and examined the cytotoxicity of the extract and its nine fractions to HepG2 cells. The contribution of all identified products from the 23 groups to the chemicals in the nine fractions was listed in "products identified in lab-CDs" worksheet in Supplementary Excel File. As shown in Supplementary Fig. 8, the extract as a whole exhibited similar cytotoxicity to the raw photodegradation products (Fig. 2b, e), suggesting that most of the cytotoxic products were recovered by solid-phase extraction. Moreover, 8 of the 9 fractions showed significantly ($p < 0.05$, see Source Data file for detailed $p$ values) inhibitive effects on the viability of HepG2 cells. Therefore, the photo-induced cytotoxicity of the laboratory-synthesized CDs was a combined effect of the various degradation products in the 23 groups. Nevertheless, the contribution of Group 23 to the cytotoxicity was especially obvious based on the fact that Fraction 1 containing chemicals from most groups except Group 23 showed no significant (one-

way ANOVA, $p = 0.44$) cytotoxic effects and Fraction 9 containing only chemicals from Group 23 showed a typical dose-dependent response.

**Cytotoxicity and degradation products of other CDs.** To determine whether photodegradation-induced cytotoxicity is common to CDs or specific to our laboratory-synthesized CDs, we subjected CDs purchased from Sigma-Aldrich to similar cell viability analyses. These commercial CDs were synthesized by autoclaving citric acid and urea[56], a method different from ours. Under this condition, they possessed different physicochemical properties (e.g., larger size, more disordered structure, N-containing) (see Supplementary Fig. 9 for details), as compared to our laboratory-synthesized CDs. When tested in the same three cell lines, commercial CDs also showed irradiation-time-dependent cytotoxicity (Supplementary Fig. 10a–c). The <3 kD fraction obtained after 8 days of irradiation was also the key contributor to the cytotoxicity of the irradiated commercial CDs (Supplementary Fig. 10d–f). Degradation kinetics of commercial CDs were similar to the laboratory-synthesized CDs, with ~63% photolyzed into a < 3 kD fraction after 8 days of irradiation (Supplementary Fig. 11).

Among the degradation products of the irradiated commercial CDs, 123 substances (see "products identified in both CDs" worksheet in Supplementary Excel File) associated with cytotoxicity were also detected in our laboratory-synthesized CDs. Among these 123 products, a chemical formula could be assigned to 74 and a substructure to 22 substances. The structures of the remaining products could not be identified due to low-intensity signals or unavailable MS/MS spectra. Based on the formulas of the 74 products, 51 (68.9%) had a degree of unsaturation >3 and 31 (41.9%) an unsaturation degree >6. The formulas of two products were $C_{28}H_{22}$ and $C_{31}H_{20}$; their elemental composition and high degree of unsaturation suggested that they were polycyclic aromatic hydrocarbons. The 22 products with identified substructures (red colored in "products identified in both CDs" worksheet of Supplementary Excel File) included all products in Group 23 as well as 6 PEG-conjugated products in Group 22 and Group 9 that were also obtained from the laboratory-synthesized CDs. The resemblances in the chemical structures of the photodegradation products explain the similarity in the cytotoxicity of these two types of CDs.

Besides the CDs from Sigma-Aldrich, we also prepared another three CDs, two nitrogen doped ($CD_{N1}$ and $CD_{N2}$) and one silicon doped ($CD_{Si}$). They were synthesized by microwaving glycerol and ethanolamine[57], diethanolamine[58], or glycerol and (3-aminopropyl)-triethoxysilane[59]. Detailed characterization results were shown in Supplementary Fig. 12. More importantly, all three CDs could degrade under the illumination of white fluorescent light with 22.0–37.2% photolyzed into <3 kD molecules after 8 days of irradiation (Supplementary Fig. 13a). The photodegradation products also exhibited remarkable cytotoxicity to HepG2 cells (Supplementary Fig. 13b), suggesting that the photo-induced cytotoxicity might be common to CDs regardless of their composition.

Overall, we have demonstrated photodegradation-induced cytotoxicity in normal and malignant human cells exposed to laboratory-synthesized and commercial CDs from Sigma-Aldrich that have been irradiated with white fluorescent light. Through RHT analysis, major DEGs detected were associated with cancer and biological pathways involved in the cytotoxicity include inflammation, endocrine disruption, xenobiotics metabolism, DNA damage, and the immune response. Cytotoxicity was mainly attributed to photolyzed products in the <3 kD fraction separated from irradiated CDs. Using a nontarget QTOF method,

we identified the formulas and structures of these products. The photolyzed products of both types of CDs were found to include 123 substances that correlated with the cytotoxicity in at least one of the three tested cell lines. Similar photodegradation kinetics and photo-induced cytotoxicity were also observed on another three N- or Si-doped CDs. Although CDs have been shown to be biocompatible under dark conditions, our results demonstrate that CDs can generate cytotoxic substances under light-irradiated conditions. Future evaluation of their safety should therefore, consider light exposure or other conditions depending on the application.

## Methods

**CD preparation and characterization.** The CDs used in the first part of this study (referred to herein as laboratory-synthesized CDs) were synthesized in our laboratory by the microwave pyrolysis of a mixture of 5 mL of PEG, 1.5 mL of water, and 1.1 g of glucose[18]. They were then purified by diafiltration through a 3 kD regenerated cellulose membrane in a Millipore stirred cell. The morphology of the purified CDs was visualized by TEM (JEM-200CX from JEOL, Tokyo, Japan) and AFM (SPM-9700, Shimadzu, Kyoto, Japan). Their size distribution in the experimental medium was measured using a dynamic light scattering particle sizer (ZetaPALS from Brookhaven Instruments, New York, USA). Further physicochemical characterization of the CDs included their UV–Vis absorption profile (UV-2450; Shimadzu, Kyoto, Japan), elemental composition (Vario MICRO CHN element analyzer; Elementar Analysen Systeme GmbH, Germany), photoluminescence (F-7000; Hitachi, Tokyo, Japan), and XRD (ARL X'TRA; Thermo-Fisher Scientific, Massachusetts, USA). They were also analyzed using FTIR (Tensor27; Bruker, Massachusetts, America), Raman (Labram HR800; Horiba Jobin-Yvon, Paris, France), and XPS (PHI 5000 VersaProbe II; Ulvac-Phi, Kanagawa, Japan) spectroscopy. Similar characterization was also performed for the commercial CDs from Sigma-Aldrich as well as for $CD_{N1}$, $CD_{N2}$, and $CD_{Si}$.

**Photodegradation kinetics experiments.** Potential photodegradation of the laboratory-synthesized CDs was examined by exposing 300 mg carbon/L of CDs in triplicate to white fluorescent light with different intensity (0, 15, 30, 60, and 90 μmol photons/m²/s) in a plant incubator for 0, 0.5, 1, 4, and 8 days. At each time point, 10 mL of the CDs was collected from each replicate, ultrafiltered through a 3 kD regenerated cellulose membrane, and the total organic carbon concentration in the <3 kD fraction was measured by an Elementar vario TOC analyzer (Elementar Analysen systeme GmbH, Germany). Potential effects of light wavelength (390, 450, 520, and 620 nm), temperature (18, 30, 35, and 40 °C), pH (5, 6, 7, 8, and 9), ionic strength (0, 4.6, 13.8, 41.5, and 124.4 mM), and particle size (2.5 and 4.8 nm) on the photodegradation of the CDs were also examined following a similar procedure to that of light intensity. The particle size was manipulated by varying the microwaving time during CD synthesis.

To reveal the mechanisms underlying the photodegradation of CDs, potential production of reactive free radical species by the laboratory-synthesized CDs (300 mg carbon/L) exposed to white fluorescent light (60 μmol photons/m²/s) was examined. Two control treatments with the same concentration of CDs in the dark or with the addition of the <3 kD photodegradation products of CDs only were also included to ensure that the radical signals observed were photo-induced and resulted from the CD particles themselves (i.e., not from the photodegradation products). Each replicate (except those of the control treatment in the dark) was irradiated with white fluorescent light for 10 min in the presence of 200 mM DMPO or 10 mM TEMP and then the radicals were monitored by EPR (Bruker E500, Karlsruhe, Germany). EPR settings were as follows: center field 3503 G, sweep width 150 G, sweep time 40 s, 3 scans per sample, modulation frequency 100 kHz, modulation amplitude 1.0 G, microwave frequency 9.83 GHz, and microwave power 10.02 mW. The signals thus obtained were processed by Xepr software to identify the radical species. To further verify the roles of radicals, photodegradation kinetics of the laboratory-synthesized CDs with and without the presence of the hydroxyl radical scavenger isopropanol (10 mM) was compared following a similar procedure to the degradation experiments above. Additionally, degradation of the laboratory-synthesized CDs in the dark with and without the presence of the hydroxyl radial was also compared. The hydroxyl radical was produced by the Fenton reaction through the addition of $H_2O_2$ (0.3, 1, and 3 mM) and $FeCl_2$ (0.3, 1, and 3 mM) in the molar ratio of 1:1. Control treatments without any addition of $H_2O_2$ and $FeCl_2$ or with the addition of $H_2O_2$ (0.3, 1, and 3 mM) only were also included to exclude the potential oxidation effects of $H_2O_2$ itself. As the Fenton reaction was completed within minutes, this degradation experiment only lasted 1 h. The cumulative production of hydroxyl radicals during this period was also determined for each treatment following the method as described in Louit et al.[60].

**Cell viability assay.** The effect on human cell viability of the laboratory-synthesized CDs exposed to white fluorescent light (60 μmol photons/m²/s) for 0, 0.5, 1, 4, and 8 days was investigated in HeLa, HepG2, and HEK-293 cells. All three cell lines were obtained from Shanghai Zhongqiao-Xinzhou Biotechnology Co.,

Ltd. (Shanghai, China). They were cultured in DMEM (KeyGEN BioTECH Co., Ltd., Nanjing, China) containing 10% fetal bovine serum (FBS, ScienCell, USA) and incubated at a temperature of 37 °C in a humidified atmosphere of 5% $CO_2$. The cell viability assay kit (CCK-8) was purchased from Nanjing Yifeixue Biotechnology Co., Ltd. (Nanjing, China). For the viability assays, cells from the three cell lines were cultured in a 96-well plate at a density of ~10,000 cells per well. After a 24-h incubation, the culture medium was decanted and the cells were exposed to different concentrations of CDs (0, 10, 30, 100, and 300 mg carbon/L, six replicates each) for 24 h, after which the medium was replaced with 100 μL of DMEM containing 10% FBS and 10 μL of CCK-8 reagent. After a 3-h incubation, the optical density (OD) of the cultures was measured at a wavelength of 450 nm using a ThermoFisher Scientific Varioskan Flash multifunctional microplate reader. The relative cell viability was calculated according to Eq. (1):

$$\text{Relative cell viability } (\%) = \frac{OD_{Treated} - OD_{Blank}}{OD_{Control} - OD_{Blank}} \times 100\%, \tag{1}$$

where $OD_{Control}$ and $OD_{Treated}$ represent the OD value of the culture in the absence and presence of CDs, respectively, and $OD_{Blank}$ is the OD value of the culture without cell inoculation.

To further investigate whether cytotoxicity originated from the CDs themselves or from the chemicals they released into the medium following their irradiation, CDs at a concentration of 900 mg carbon/L in DMEM were irradiated for 8 days and then divided into two groups. One was used directly in the viability assay at eight concentrations (0, 1, 3, 10, 30, 100, 300, and 900 mg carbon/L). The other was ultrafiltered through a 3 kD ultrafiltration membrane to separate the CDs from the chemicals they released into the medium during the 8-day irradiation. The >3 kD and <3 kD fractions were tested separately in cell viability assays as described above for the cytotoxicity tests. To determine whether the photodegradation-induced inhibition of cell viability was specific to our laboratory-synthesized CDs or a general characteristic of CDs, commercial CDs purchased from Sigma-Aldrich were subjected to similar cell viability assays. The cytotoxicity of the <3 kD photodegradation products from $CD_{N1}$, $CD_{N2}$, and $CD_{Si}$ to HepG2 cells was also examined after 8-day irradiation.

**RHT experiment.** An amplicon-seq technology was applied to determine the transcriptional expression of 1200 selected genes[32]. For this purspose, HepG2 cells ($1 \times 10^5$ cells/mL) in 12-well plates were exposed to seven fivefold dilutions of n-CD (0.019–300 mg carbon/L), $i_8$-CD (0.00013–2 mg carbon/L), $i_8$-$CD_{<3 kD}$ (0.00019–3 mg carbon/L), and $i_8$-$CD_{>3 kD}$ (0.019–300 mg carbon/L) with a single replicate. Four vehicle controls (0.1% v/v of methanol) were also included. As the cytotoxicity of n-CD and $i_8$-$CD_{>3 kD}$ was extremely low, the highest concentration, corresponding to that used in the time-related viability assay described above, was tested. By contrast, the highest concentration of $i_8$-CD and $i_8$-$CD_{<3 kD}$ was their 20% effective concentration for the inhibition of viability. After a 24 h exposure to the CD preparations, the cells were collected and their total RNA was isolated using an RNeasy mini kit (Qiagen, Hilden, Germany). The 32 RNA samples (28 treatments and 4 vehicle controls) were stored at −80 °C until further analysis. RNA concentrations in the samples were measured using a Quant-iT RNA HS assay kit. Libraries were prepared using 10 ng of RNA from each sample, the Ion AmpliSeq library kit 2.0, and Ion AmpliSeq custom panels (Thermo Fisher Scientific, Waltham, MA), followed by high-throughput sequencing of the RHT panel on an Ion Torrent Proton (Thermo Fisher Scientific, Waltham, MA).

A dose–response analysis of each gene was then conducted[32,61]. The fold change in the expression of genes that indicated a significant ($p < 0.05$, Akaike information criterion) concentration-dependent pattern was calculated for each treatment by dividing the respective sequence counts by the mean count of the vehicle control. The log2-transformed fold change in the expression of each gene vs. the log10-transformed CD concentration was fitted with nine concentration-effect models representing three curve types (sigmoidal, linear, and U-shaped) in R 3.6.1 (Supplementary Table 1). DEGs were the genes whose best-fit model showed a significant curve-fitting performance ($p < 0.05$, Akaike information criterion). The $EC_{DEG}$ was obtained by the best-fit model. Thus, $EC_{DEG}$ was defined as the concentration causing a 50% maximum effect for models with a sigmoidal-type curve, a 1.5-fold change for models with a linear curve type, or a 1.5-fold change within the first monotonic portion of models with a U-shaped curve type. Based on the $EC_{DEG}$, the effective concentration with respect to the GO terms ($EC_{GO}$) or the KEGG pathways ($EC_{KEGG}$) related to the DEGs identified above was further derived by the geometric mean of the $EC_{DEG}$. Only GO terms or KEGG pathways with at least three matched DEGs were included in the analysis due to the number limit for the calculation of the mean and standard deviation. The overall biological potency of each sample was characterized by fitting the proportionally ranked $EC_{DEG}$, $EC_{GO}$, and $EC_{KEGG}$ values into a probability cumulative distribution curve of log-normal distribution in R 3.6.1.

Besides the modes of toxic action, potential structural information about the toxicants was obtained based on the associations between chemicals and genes/pathways available in the CTD. According to the chemical-gene interactions of *Homo sapiens* available in the database, substances able to activate co-detected DEGs in $i_8$-CD and $i_8$-$CD_{<3 kD}$ were identified and visualized by Cytoscape 3.7.2. Substances that disturbed co-enriched GO or KEGG pathways in $i_8$-CD and $i_8$-$CD_{<3 kD}$ were identified as well, based on the associations between chemicals

and GO or KEGG pathways. The substances thus acquired were further screened according to the criterion that they had to have affected >50% of the co-detected DEGs or GO/KEGG pathways. The structures of these substances were then determined.

**Identification of the degradation products**. Photodegradation products were enriched by solid-phase extraction. Briefly, 300 mg carbon/L of laboratory-synthesized CDs were irradiated with white fluorescent light (60 μmol photons/m$^2$/s) for 0, 0.5, 1, 4, or 8 days and the suspensions were then ultrafiltered through a Millipore 3 kD membrane. The ultrafiltrate was passed through an Oasis HLB cartridge (500 mg, 6 mL, Waters, USA) pre-cleaned with methanol and Milli-Q water. The photodegradation products were eluted using 5 mL of methanol and concentrated to 1 mL by nitrogen blow down. Five microliter of the concentrate was injected into a HPLC apparatus (Infinity 1260, Agilent Technologies, Wald-bronn, Germany) coupled with a high-resolution QTOF mass spectrometer (Triple TOF 5600, AB Sciex, Foster City, CA). The electrospray ionization source was operated in positive ion mode. The obtained compounds were separated on a BEH C18 column (2.1 mm × 50 mm, 2.5 μm, Waters, U.S.) at 40 °C. The mobile phase was 5% acetonitrile in water (A) and methanol (B) with a flow rate of 400 μL/min. The HPLC gradient was as follows: 100% solvent A for 1 min, decreased to 0% solvent A until 29 min and held until 32 min, returned to 100% solvent A until 32.1 min and held at 100% solvent A until 38 min for equilibration. An information-dependent acquisition was conducted with one TOF-MS analysis (m/z 100–1250) and 20 dependent TOF-MS/MS analyses (m/z 50–1250) per cycle in Analyst TF 1.5. Additional experimental parameters were: nebulizer gas, 55 psi; heater gas, 55 psi; curtain gas, 35 psi; temperature, 550 °C; ionspray voltage floating, 4500 V; declustering potential, 100 V. The collision energy and collision energy spread for the TOF-MS/MS analysis was 40 V and 20 V, respectively. All gases used in the analysis were nitrogen.

The peak features of all samples as obtained from the HPLC-QTOF analysis were extracted using Peakview 1.2 software (AB Sciex, Foster City, CA). Photodegradation products were identified based on a peak intensity three times higher than that of the control treatment (i.e., the <3 kD ultrafiltrate of n-CD). Similarities in the formation kinetics of the different photodegradation products were compared in a two-tailed Pearson correlation analysis (SPSS 23.0). Only the products detected at all time points were included and their peak areas were normalized with Z-score standardization. Molecular network and community analyses were performed in Gephi (0.9.2) to visualize similarities in the formation kinetics curves[53]. The photodegradation products were divided into different communities using the Louvain method[62]. The pair of products with p < 0.001 was imported to Gephi and a layout algorithm (Fruchterman Reingold) was chosen. To identify products or communities associated with toxicity, the correlation analysis between the inhibition of cell viability by CDs irradiated for different durations and the time-related change in the peak area of each product or community was also analyzed, using SPSS 23.0. Similar to the correlation analysis of the formation kinetics, the cytotoxicity data were expressed as a vector including cell viability inhibition by CDs irradiated for different durations, and the peak area of each photodegradation product was also expressed as a vector depending on the irradiation time of CDs. Before correlation analysis, the toxicity data and peak area were log10-transformed and further normalized with Z-score standardization. The peak area of the community was the average peak area (normalized) of all photodegradation products in the corresponding community. The products or communities associated with toxicity were identified also by the two-tailed Pearson correlation analysis. In addition, the product-cell pair with a significantly (p < 0.05) positive correlation was also imported to the molecular network of the formation kinetics (Gephi 0.9.2) to visualize the association between the photodegradation products and cell lines. To identify the chemical structures of the products positively associated with cytotoxicity, their potential molecular formulas were derived based on the monoisotopic mass and the isotopic distribution results in the TOF-MS spectra[63]. The structures represented by the formulas were then postulated based on the fragment ions in the TOF-MS/MS spectra. Each structure was assigned a confidence level according to the criteria reported in Schymanski et al.[64]. Namely, five levels (i.e., level 1: confirmed structure, level 2: probable structure, level 3: tentative candidates, level 4: unequivocal molecular formula, level 5: exact mass of interest) were proposed with the identification confidence in the order from high to low. The structures were identified at levels 1–3 herein. To annotate structural information for the photodegradation products representing fewer fragment ions or without any fragment ions in the MS/MS spectra, a homologue and high-frequency fragment analysis was conducted in Matlab R2016a[65]. Similar to the laboratory-synthesized CDs, the degradation products of the commercial CDs from Sigma-Aldrich were analyzed using the same high-throughput nontarget method but for 8-day irradiated samples only.

**Reporting summary**. Further information on research design is available in the Nature Research Reporting Summary linked to this article.

## Data availability

The data supporting the findings of this study are available within this article and its Supplementary Information Files. All other relevant data are available from the corresponding authors upon reasonable request. Source data are provided with this paper.

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

## Acknowledgements
We appreciate Prof. Shu-Juan Zhang's helpful discussion about the photodegradation mechanisms. This work was supported by the National Natural Science Foundation of China (21822605, 22022603, and 21677068), the Fundamental Research Funds for the Central Universities (021114380148), Chinese Public Science and Technology Research Funds for Ocean Projects (201505034), and the program B for Outstanding Ph.D. candidate of Nanjing University.

## Author contributions
A.J.M. conceived the idea, with S.W. and X.W.Z. providing additional input. Y.Y.L. synthesized the CDs, did the characterization, and performed the photodegradation kinetics and cell viability experiments. N.Y.Y., R.J., and S.W. identified the photo-degradation products. W.D.F., L.Y.Y., and X.W.Z. performed the RHT experiment as well as the cell invasion and transformation assays, and analyzed the transcriptome data. Q.G.T. analyzed the data and drew the figures. Y.Y.L., W.D.F., and N.Y.Y. wrote the paper with contribution from all co-authors.

## Competing interests
The authors declare no competing interests.
