## [Peer Review File · Nature Communications]

Reviewers' comments:

Reviewer #1 (Remarks to the Author):

Manuscript by Liu et al. describes an important phenomenon of photo-induced cytotoxicity of carbon dots. With respect to a big application potential of CDs in photocatalysis, the obtained results would be of a possible general interest for a broad community.

However, in the current state, the manuscript is not acceptable for publication in Nature Communications. Despite of a quite complex analysis of prepared CDs and degradation products, there are several issues related to the inconclusive generalization of the observed cytotoxicity mechanisms.

Before possible further consideration of the article, the authors should address following aspects:

- i) What about possible effect of carbon dot particle size on photodegradation process and secondary cytotoxicity?
- ii) If the authors want to generalize the observed phenomenon also for other types of CDs they have to perform similarly precise chemical/structural/morphological analysis of commercial carbon dots. Indeed, it looks like both types of studied dots would contain different functional groups anchored onto the surface. Also, there are well expectable differences in particle size distribution. So, it is really unproved conclusion that the degradation products come mainly from graphitic core.
- iii) Many types of carbon dots contain nitrogen embedded in a carbogenic core or as a part of N-bearing surface functional groups. Such systems should be tested in the case the authors aim to generalize the photo-induced cytotoxicity of CDs.
- iv) What about the effect of the light wavelength and energy??
- v) The authors identified many species responsible for the final cytotoxicity. Some specification/identification of the most important cytotoxic "players" would help to sell the story for the readers.

Reviewer #2 (Remarks to the Author):

In this study, Liu et al proposed a study to examine the degradation of carbon dots under long term irradiation as well as the subsequent toxic effects by degradation products. While some literature reports have demonstrated that the analogues (graphene oxide, graphene, graphene quantum dots, etc.) of carbon dots are able to degrade into small molecules (Bai et al, Phys. Chem. C, 2014, 10519-10529; Mukherjee et al, Nanoscale, 2018, 1180–1188; Kurapati et al, 2018, 11722-11727). it's not surprising that carbon dots could be degraded under radiation. Based on a HPLC-QTOF analysis, more than 1800 features were identified. A kinetic correlation analysis revealed 655 features correlated with cytotoxicity of CD. However, the authors did not provide further validation on whether some of these identified features are really responsible for cytotoxicity. Besides, the authors claimed the carcinogenic effects of CDs based on transcriptome analysis. Since there are no further biological experiments to validate this effect, I think it's premature to make this claim. Animal experiments must be also performed to validate the carcinogenic effects. In terms of experimental description, the information provided in current manuscript is insufficient for repeatable operation by other researchers. Overall, this study showed some intriguing phenomenon on the toxicity of CDs. However, the authors failed to provide strong evidences to support these findings. Following are my specific comments:

Major comments:

1. A detailed experimental description of photodegradation and correlation analysis between degradation products and cytotoxicity should be provided.
2. How do the authors demonstrate that the TOCs were resulting from CD degradation rather than the absorbed small molecules during CD synthesis?
3. I would suggest the authors to collect the degradation products with high correlation with cytotoxicity for further validation.
4. Traditional biological experiments should be performed to validate carcinogenic effects.
5. Animal experiments should also be designed to validate cell effects.
6. Will the degradation of CDs depend on radiation intensity?

7. What's the mechanism involved in the degradation of CDs?
8. Figure 2 shows that only CDs at a long time radiation treatment (>1d) have toxicity at extremely high doses (>100 mg/L). How do the authors justify the radiation treatment and doses in terms of real exposure scenarios?
9. TEM is not an appropriate method to visualize CDs, and Figure 1a has poor quality. AFM should be performed.
10. Since the synthesis of CDs has been well reported, more experiments should be performed on the degradation of CDs to replace the characterization data of XPS, XRD, etc. Will the pH, ionic strength, temperature have effects on CD degradation.

Minor comments:

1. Abbreviations should be defined during their first use, such as HPLC-QTOF
2. Only the authors made substantial contributions in the manuscript should be listed as first author or corresponding author. There are too many equal contributions in the manuscript.
3. What kind of instrument provides the white fluorescent light for degradation experiments?
4. Line 62, delete "were".
5. The title of table 1 should be concise.
6. Figure 4b and 4d should be removed. Although these two images looks good, I failed to acquire sufficient information from these images.

Changes made in response to Reviewers' comments **(Italicized—reviewers' comments)**

Reviewer #1:

Comment 1: What about possible effect of carbon dot particle size on photodegradation process and secondary cytotoxicity?

Response: Thanks. Following this reviewer's suggestion, we synthesized CDs with different sizes (2.5 and 4.8 nm) by regulating the microwaving time during CD synthesis. Despite the difference in particle size, similar photodegradation kinetics was observed (Fig. 1f). As photodegradation had a dominant contribution to the cytotoxicity of CDs, the negligible effects of particle size on photodegradation further indicate the negligible effects of size on the secondary cytotoxicity of CDs. This has now been clarified in Line 129-131.

Comment 2: If the authors want to generalize the observed phenomenon also for other types of CDs they have to perform similarly precise chemical/structural/morphological analysis of commercial carbon dots. Indeed, it looks like both types of studied dots would contain different functional groups anchored onto the surface. Also, there are well expectable differences in particle size distribution. So, it is really unproved conclusion that the degradation products come mainly from graphitic core.

Response: Thanks. Following this reviewer's suggestion, we have now provided the detailed characterization results of the commercial CDs from Sigma-Aldrich. As shown in Supplementary Fig. S7, the commercial CDs possessed different physicochemical properties (*e.g.*, larger size, more disordered structure, N-containing) (see Supplementary Fig. S7 for details), as compared to our laboratory-synthesized CDs, which has now been clarified in Line 359-361.

As for the comment whether the degradation products come mainly from graphitic core, we have now removed the misleading speculation that 'Since the surface coating of the commercial CDs is likely different from our CDs, these 126 shared substances probably originated from

the CD cores and had a major contribution to the photo-induced cytotoxicity of both CDs' because we have no convincing evidence to support this hypothesis. Based on the photodegradation kinetics results we have now provided in Fig. 1, a maximum of 97.1% of the CDs degraded (*i.e.*, CDs could degrade completely). It implies that the photodegradation products came not only from the surface layer but also from the inner parts of CDs. As the cytotoxicity of CDs increased with irradiation time, it indicates that not only the photodegradation products from the surface layer but also those from the core are cytotoxic.

Comment 3: Many types of carbon dots contain nitrogen embedded in a carbogenic core or as a part of N-bearing surface functional groups. Such systems should be tested in the case the authors aim to generalize the photo-induced cytotoxicity of CDs.

Response: Thanks. Following this reviewer's suggestion, we have now prepared another three types of CDs, two N-doped (CD_{N1} and CD_{N2}) and the other one Si-doped (CD_{Si}). Based on the characterization results in Supplementary Fig. S10, the sizes of these three CDs were in the range of 3.1-9.4 nm with significant photoluminescence. XPS results reveal the existence of C, N, and O in CD_{N1} and CD_{N2} as well as the presence of C, N, O, and Si in CD_{Si} . Then we investigated the photodegradation kinetics of these three types of CDs under the irradiation of white fluorescent light. As shown in Supplementary Fig. S11a, a significant time-dependent photodegradation of all three CDs was observed. Moreover, these photodegradation products showed significant cytotoxicity to HepG2 cells (Supplementary Fig. S11b). These results suggested that the findings of our study can be extended to other CDs. This information has now been added in Line 382-390.

Comment 4: What about the effect of the light wavelength and energy?

Response: Thanks. Following this reviewer's suggestion, we have now examined the effects of light wavelength and intensity as well as other environmental factors (*e.g.*, temperature, pH, ionic strength) on CD photodegradation. As shown in Fig. 1, CD degradation increased with light intensity. Negligible degradation occurred in the dark while 19.1%, 25.7%, 34.0%, and

42.8% of the CDs degraded into dissolved molecules < 3 kD in size after 8 days of irradiation to white fluorescent light with the intensity of 15, 30, 60, and 90 $\mu\text{mol photons/m}^2/\text{s}$, respectively. As for wavelength, CD degradation increased as the light wavelength decreased from 620 nm (red) to 450 nm (blue) and remained constant thereafter (Fig. 1b). In this case, nearly all CDs (97.1–96.9%) degraded after 8 days of irradiation to blue (450 nm) and purple (390 nm) light. Additionally, more CDs degraded at higher temperature (18–40 °C, Fig. 1c) or pH (5–9, Fig. 1d). By contrast, ionic strength (0–124.4 mM, Fig. 1e) and particle size (2.5 and 4.8 nm, Fig. 1f) had no significant ($p > 0.05$) effects on CD degradation. This has now been clarified in Line 119-131.

Besides the effects of various environmental factors on CD degradation, we also investigated the photodegradation mechanisms of CDs. For this purpose, we used electron paramagnetic resonance spectroscopy (EPR) in combination with two spin traps 5,5-dimethyl-1-pyrroline-N-oxide (DMPO) and 2,2,6,6-tetramethylpiperidine (TEMP) to detect the production of reactive free radical species during degradation. As shown in Fig. 1g, both hydroxyl and alkyl radicals but no singlet oxygen were formed when the CDs were irradiated with white fluorescent light for 10 min. Contrastively, no radicals were detected in the dark or in the medium with the addition of < 3 kD molecules degraded from the CDs only. These phenomena imply that the free radical species were produced by the CDs themselves under irradiation. To further explore how these radicals might be involved in CD photodegradation, we examined the degradation kinetics of the CDs in the presence of a hydroxyl radical scavenger (*i.e.*, isopropanol). As shown in Fig. 1h, CD degradation decreased significantly ($p < 0.05$) in the presence of isopropanol, suggesting the importance of hydroxyl radicals in CD photodegradation. Similar experiment cannot be performed for alkyl radicals as no effective scavenger is available due to their complicated speciation. Nevertheless, since isopropanol cannot completely prevent CD photodegradation, alkyl or other radicals may also play important roles. This has now been clarified in Line 132-147.

Comment 5: The authors identified many species responsible for the final cytotoxicity. Some

specification/identification of the most important cytotoxic “players” would help to sell the story for the readers.

Response: Thanks so much for the suggestion. To explore the main cytotoxicity “players” of the photodegradation products, we concentrated the degradation products by solid-phase extraction, divided them into 9 fractions based on their retention time in HPLC, and then examined the cytotoxicity of each fraction to HepG2 cells. As shown in Supplementary Fig. S6, the extract as a whole exhibited similar cytotoxicity to the raw photodegradation products (Fig. 2b,e), suggesting that most of the cytotoxic products were recovered by solid-phase extraction. Moreover, 8 of the 9 fractions showed significantly ($p < 0.05$) inhibitive effects on the viability of HepG2 cells. Therefore, the photo-induced cytotoxicity of the laboratory-synthesized CDs was a combined effect of the various degradation products in the 8 fractions. Nevertheless, the contribution of Group 23 to the cytotoxicity was especially obvious based on the fact that Fraction 1 containing chemicals from most groups except Group 23 showed no significant ($p > 0.05$) cytotoxic effects and Fraction 9 containing only chemicals from Group 23 showed a typical dose-dependent response. This has now been clarified in Line 339-354.

Reviewer #2:

Comment 1: Based on a HPLC-QTOF analysis, more than 1800 features were identified. A kinetic correlation analysis revealed 655 features correlated with cytotoxicity of CD. However, the authors did not provide further validation on whether some of these identified features are really responsible for cytotoxicity.

I would suggest the authors to collect the degradation products with high correlation with cytotoxicity for further validation.

Response: As mentioned in our response to comment 5 of Reviewer #1, to explore the main cytotoxicity “players” of the photodegradation products, we concentrated the degradation products by solid-phase extraction, divided them into 9 fractions based on their retention time in HPLC, and then examined the cytotoxicity of each fraction to HepG2 cells. As shown in Supplementary Fig. S6, the extract as a whole exhibited similar cytotoxicity to the raw photodegradation products (Fig. 2b,e), suggesting that most of the cytotoxic products were recovered by solid-phase extraction. Moreover, 8 of the 9 fractions showed significantly ($p < 0.05$) inhibitive effects on the viability of HepG2 cells. Therefore, the photo-induced cytotoxicity of the laboratory-synthesized CDs was a combined effect of the various degradation products in the 8 fractions. Nevertheless, the contribution of Group 23 to the cytotoxicity was especially obvious based on the fact that Fraction 1 containing chemicals from most groups except Group 23 showed no significant ($p > 0.05$) cytotoxic effects and Fraction 9 containing only chemicals from Group 23 showed a typical dose-dependent response. This has now been clarified in Line 339-354.

Comment 2: the authors claimed the carcinogenic effects of CDs based on transcriptome analysis. Since there are no further biological experiments to validate this effect, I think it's premature to make this claim. Animal experiments must be also performed to validate the carcinogenic effects.

Traditional biological experiments should be performed to validate carcinogenic effects.

Animal experiments should also be designed to validate cell effects.

Overall, this study showed some intriguing phenomenon on the toxicity of CDs. However, the authors failed to provide strong evidences to support these findings.

Response: Thanks for the comment. We agree with this reviewer that it's premature to claim that the photodegradation products of CDs are carcinogenic based on transcriptome results only. To further examine the potential carcinogenic effects of the photodegradation products, we performed another two experiments [*i.e.*, cell invasion and transformation (as indicators of carcinogenicity) experiments]. Detailed information how these two assays were performed has now been provided in Supplementary Information. As shown in Supplementary Fig. S3, both i₈-CD and i₈-CD_{<3kD} showed significantly ($p < 0.05$) inductive effects on the invasion and transformation of HepG2 cells especially in the two lower concentration treatments while no obvious effects were observed for n-CD and i₈-CD_{<3kD}. The highest concentrations of i₈-CD and i₈-CD_{<3kD} used herein were their 20% effective concentration for the inhibition of cell viability. The significant cytotoxicity of the highest concentration treatments of i₈-CD and i₈-CD_{<3kD} explains their much less inductive effects on cell invasion and transformation as compared to the other two lower concentration treatments. These results further support our hypothesis that the photodegradation products might be carcinogenic. This has now been clarified in Line 226-236. Nevertheless, animal experiments MUST be performed to finally prove this hypothesis, which is not the objective of the present study. The main story we're planning to tell is 1) CDs were photodegradable, their degradation kinetics were affected by several environmental factors, and the photo-induced production of free radicals plays important roles in CD degradation; 2) the photodegradation products of CDs were cytotoxic and various biological pathways were affected; 3) 1431 degradation products were identified by high-throughput, non-target HPLC-QTOF and 499 of them were cytotoxicity-related. Carcinogenicity was just one of the possible toxic modes of action of the photodegradation products, which could be examined in future studies.

Comment 3: In terms of experimental description, the information provided in current manuscript is insufficient for repeatable operation by other researchers.

A detailed experimental description of photodegradation and correlation analysis between degradation products and cytotoxicity should be provided.

Response: Thanks for the suggestion. Now detailed information how the different experiments were performed has now been clarified in Line 421-450, 479-480, and 547-584 as well as in Supplementary Information (cell invasion and transformation assays, fractionation of the degradation products and cell viability assay). Detailed experimental description of photodegradation and correlation analysis between degradation products and cytotoxicity has also been provided in Line 547-584 (yellow highlighted).

Comment 4: How do the authors demonstrate that the TOCs were resulting from CD degradation rather than the absorbed small molecules during CD synthesis?

Response: In the present study, right before the photodegradation experiment, a certain volume of CD suspension (*e.g.*, 400 mL) was put in a stirred cell (Millipore) with a 3-kD regenerated cellulose membrane at the bottom. Then the suspension was pressed by high-pressure nitrogen gas and any molecules < 3 kD would pass through the membrane together with water, while the CD particles were retained on the membrane. When the liquid volume was reduced to 40 mL, the ultrafiltration was stopped, the CDs retained in the stirred cell were diluted back to 400 mL, and then the ultrafiltration process was repeated until the TOC content in the ultrafiltrate was comparable to the background. Through this repeated ultrafiltration, any small organic molecules retained in the medium of the CD suspension was removed. As for the small organic molecules potentially adsorbed on the CD surface, they had negligible contribution to the TOC content of the < 3 kD ultrafiltrate as measured at different time points of the photodegradation kinetics experiments. This conclusion was supported by our finding that no significant degradation of CDs was observed in the dark (Fig. 1a). In other words, if there's significant desorption of small organic molecules from the CDs during the photodegradation period, this should also happen in the dark, which is obviously not the case in the present study. Moreover, when we examined the photodegradation kinetics of CDs under different environmental conditions, a maximum of 97.1% of CDs (*i.e.*, nearly all CDs) degraded after 8-d irradiation

(Fig. 1b). All these phenomena imply that photodegradation of CDs is real and surface desorption of the small organic molecules (if any) can be neglected.

Comment 5: Since the synthesis of CDs has been well reported, more experiments should be performed on the degradation of CDs to replace the characterization data of XPS, XRD, etc. Will the pH, ionic strength, temperature have effects on CD degradation.

Will the degradation of CDs depend on radiation intensity? What's the mechanism involved in the degradation of CDs?

Response: Thanks. Following this reviewer's suggestion, we have now examined the effects of light wavelength and intensity as well as other environmental factors (*e.g.*, temperature, pH, ionic strength) on CD photodegradation. As shown in Fig. 1, CD degradation increased with light intensity. Negligible degradation occurred in the dark while 19.1%, 25.7%, 34.0%, and 42.8% of the CDs degraded into dissolved molecules < 3 kD in size after 8 days of irradiation to white fluorescent light with the intensity of 15, 30, 60, and 90 $\mu\text{mol photons/m}^2/\text{s}$, respectively. As for wavelength, CD degradation increased as the light wavelength decreased from 620 nm (red) to 450 nm (blue) and remained constant thereafter (Fig. 1b). In this case, nearly all CDs (97.1–96.9%) degraded after 8 days of irradiation to blue (450 nm) and purple (390 nm) light. Additionally, more CDs degraded at higher temperature (18–40 °C, Fig. 1c) or pH (5–9, Fig. 1d). By contrast, ionic strength (0–124.4 mM, Fig. 1e) and particle size (2.5 and 4.8 nm, Fig. 1f) had no significant ($p > 0.05$) effects on CD degradation. This has now been clarified in Line 119-131.

Besides the effects of various environmental factors on CD degradation, we also investigated the photodegradation mechanisms of CDs. For this purpose, we used electron paramagnetic resonance spectroscopy (EPR) in combination with two spin traps 5,5-dimethyl-1-pyrroline-N-oxide (DMPO) and 2,2,6,6-tetramethylpiperidine (TEMP) to detect the production of reactive free radical species during degradation. As shown in Fig. 1g, both hydroxyl and alkyl radicals but no singlet oxygen were formed when the CDs were irradiated with white fluorescent light for 10 min. Contrastively, no radicals were detected in the dark or in the

medium with the addition of < 3 kD molecules degraded from the CDs only. These phenomena imply that the free radical species were produced by the CDs themselves under irradiation. To further explore how these radicals might be involved in CD photodegradation, we examined the degradation kinetics of the CDs in the presence of a hydroxyl radical scavenger (*i.e.*, isopropanol). As shown in Fig. 1h, CD degradation decreased significantly ($p < 0.05$) in the presence of isopropanol, suggesting the importance of hydroxyl radicals in CD photodegradation. Similar experiment cannot be performed for alkyl radicals as no effective scavenger is available due to their complicated speciation. Nevertheless, since isopropanol cannot completely prevent CD photodegradation, alkyl or other radicals may also play important roles. This has now been clarified in Line 132-147.

As for the characterization data, we'd like to keep these results as they are important to prove that the CDs we synthesized shared similar properties as those CDs reported in the literature. Therefore, the photodegradation results observed for the glucose-based CDs may be extended to other CDs, which was also verified by several other types of CDs in the present study. Nevertheless, these results have now been moved to Fig. S1 in Supplementary Information. The characterization results of the commercial CDs from Sigma-Aldrich and three other N- or Si-doped CDs were also shown in Supplementary Fig. S7 and S10, following the suggestion of Reviewer #1.

Comment 6: Figure 2 shows that only CDs at a long time radiation treatment (>1d) have toxicity at extremely high doses (>100 mg/L). How do the authors justify the radiation treatment and doses in terms of real exposure scenarios?

Response: Thanks. We appreciate this reviewer's concern about the effects of CDs on cell viability at a relatively high dose or after relatively longer-time radiation. However, cell viability based on the CCK-8 assay is not a sensitive toxicity endpoint as compared to other endpoints like those through RHT analysis used in the present study. Through the RHT experiment, we obtained the effective concentration of DEG (EC_{DEG}), GO (EC_{GO}), and KEGG (EC_{KEGG}) for n-CD, i8-CD, i8-CD_{<3kD}, and i8-CD_{>3kD}. Detailed information how EC_{DEG} , EC_{GO} ,

and EC_{KEGG} were calculated can be found in Line 498-512. As shown in Fig. 3a-c, the EC_{DEG}, EC_{GO}, and EC_{KEGG} of i₈-CD was in the range of 0.00017–3.9 mg carbon/L, 0.003–0.19 mg carbon/L, and 0.007–0.09 mg carbon/L, respectively. Similarly, significant induction of cell invasion and transformation was observed when the i₈-CD concentration was 0.08 and 0.4 mg carbon/L (Supplementary Fig. S3). Such concentration of < 3 kD photolyzed molecules can be easily obtained within 0.5 d based on the photodegradation kinetics experimental results in Fig. 1.

According to the literature (Wang et al., *Experimental Biology and Medicine*, 2011, 236: 1231; Kong et al., *Nanoscale*, 2014, 6: 5116; Kang et al., *Scientific Reports*, 2015, 5: 11835; Song et al., *RSC Advance*, 2014, 4: 27184; Wang et al., *Current Pharmaceutical Design*, 2015, 21: 5401), the concentration of CDs used in real scenarios is in order of magnitude of mg carbon/L to g carbon/L. CDs can be irradiated intentionally during their applications in bioimaging or photocatalysis (Li et al., *Journal of Materials Chemistry*, 2012, 22: 24230; Farshbaf et al., *Artificial Cells, Nanomedicine, and Biotechnology*, 2018, 46: 1331) and they can also be irradiated unintentionally when released into the environment after application. Overall, based on the concentration of CDs used in real scenarios and their intentional or unintentional light exposure, the photo-induced cytotoxicity of CDs observed in the present study is realistic.

Comment 7: TEM is not an appropriate method to visualize CDs, and Figure 1a has poor quality. AFM should be performed.

Response: Thanks. Now the AFM images have now been provided for our laboratory-synthesized CDs as well as the other four CDs (Supplementary Fig. S1, S7, and S10). The particle size obtained from AFM was comparable to that from TEM.

*Comment 8: Abbreviations should be defined during their first use, such as HPLC-QTOF
Line 62, delete “were”.*

Response: Thanks. Revised as suggested (Line 27-28, 71-72, and 62).

Comment 9: Only the authors made substantial contributions in the manuscript should be listed as first author or corresponding author. There are too many equal contributions in the manuscript.

Response: We appreciate the worry of this reviewer about the number of first authors and corresponding authors. We always follow the criteria that only those who has substantial contributions to the manuscript can be listed as the first or corresponding authors. The experiments of this manuscript were actually completed by the students or postdoc from three research groups. As mentioned in ‘Contribution’ in Line 599-604, Ms. Yue-Yue Liu from Dr. Ai-Jun Miao’s group synthesized the CDs, did the characterization and photodegradation experiments, and performed the cell viability test. Dr. Nan-Yang Yu from the Dr. Si Wei’s group identified the photodegradation products and did the correlation analysis between the photodegradation products and cytotoxicity. Ms. Wen-Di Fang from Dr. Xiao-Wei Zhang’s group performed the RHT, cell invasion and transformation experiments, analyzed the transcriptome data. Dr. Ai-Jun Miao conceived the idea, with Dr. Si Wei and Dr. Xiao-Wei Zhang providing additional input. Therefore, the three first authors and corresponding authors really had substantial contributions to this manuscript.

Comment 10: What kind of instrument provides the white fluorescent light for degradation experiments?

Response: The photodegradation experiment was performed in a plant incubator with white fluorescent light. This information has now been provided in Line 425-426.

Comment 11: The title of table 1 should be concise.

Response: Thanks for the suggestion. Now the title of Table 1 has been simplified.

Comment 12: Figure 4b and 4d should be removed. Although these two images looks good, I failed to acquire sufficient information from these images.

Response: Thanks for the suggestion. As mentioned in Line 258-260, the intensity/area of the

peaks showed different patterns of variation with irradiation time (Fig. 4b) and the number of peaks with an m/z between 270 and 350 was larger than other peaks (Fig. 4c). As the peak area and peak number are indicators of the concentration and number of photodegradation products, these results suggest that the irradiated CDs released many small molecular substances (Fig. 4c) that had different formation kinetics (*i.e.*, different patterns of variation with irradiation time in Fig. 4b). This has now been clarified in Line 260-263.

As for Fig. 4d, we have now re-drawn this figure. Each node in Fig. 4d now represents one photodegradation product instead of one peak before [*i.e.*, a single peak is not equivalent to a single substance, which can have several peaks (adduct ions)]. Through this change, the similarity in the formation kinetics of the different photodegradation products and their correlation with cytotoxicity can be understood more easily. Therefore, totally 1431 nodes/products were detected through HPLC-QTOF. Links between nodes/products indicate their similarity in formation kinetics during the photodegradation process. Twelve communities [9 of them with significant ($p < 0.05$) correlation to cytotoxicity were numbered] indicative of the connectivity between different nodes were obtained in which > 10 nodes with similar formation kinetics clustered together. The products/nodes in the 9 communities are differently coloured (— uncorrelated to cytotoxicity, ●— correlated to the cytotoxicity in all three cells, ●— correlated to the cytotoxicity in HEK-293 and HeLa, ●— correlated to the cytotoxicity in HEK-293 and HepG2, ●— correlated to the cytotoxicity in HepG2 and HeLa, ●— correlated to the cytotoxicity in HEK-293, ●— correlated to the cytotoxicity in HeLa, ●— correlated to the cytotoxicity in HepG2) according to their correlation with the cytotoxicity in HEK-293, HeLa, and HepG2 cells. Based on the correlation intensity of each peak with cytotoxicity, three cell nodes were added, clarifying the significant ($p < 0.05$) association between the peaks in 9 of the 12 communities and the cytotoxicity in at least one cell line. From these correlation analysis, 499 products from the 9 communities were found to be associated with cytotoxicity in at least one cell line: 215 products in HepG2 cells, 415 in HeLa cells, 336 in HEK-293 cells, and 162 products that correlated with cytotoxicity in all three cell lines (Fig. 4e). Following the molecular network analysis in Fig. 4d–f, the 499 photodegradation products (nodes) associated

with cytotoxicity were subjected to a structural analysis and the structures of 212 of the 499 products were identified. Overall, we'd like to keep Fig. 4b,d, as they are the pre-requisite for Fig. 4e,f and for the structural analysis. This information has now been clarified in Line 264-291, 547-570, and in the caption of Fig. 4. Hopefully they are easier to understand now.

REVIEWER COMMENTS

Reviewer #1 (Remarks to the Author):

The authors addressed all the issues highlighted in the previous report. The reviewer appreciate the experimental works done by the authors to explain the mechanistic aspects of the study. I suggest the manuscript for publication in Nature Communications.

Reviewer #2 (Remarks to the Author):

The authors have provided sufficient data in the revised manuscript. The quality and novelty of this submission are significantly improved. It's important to see that radicals play an important role in CD degradation. This is a very interesting phenomenon, which may deserve more explorations. Following are my comments in terms of their new data and clarifications.

1. Since the degradation of CD displayed radiation and radical dependent manner, it will be convincing to simply add radicals generated by small molecules into CD solutions for assessment of degradation in dark.
2. A scheme should be added to facilitate the understanding of degradation mechanism
3. A discussion of degradation mechanisms as well as the structure-activity relationships may strengthen the fundamental implication of this study.

Changes made in response to Reviewers' comments
(Italicized—reviewers' comments)

Reviewer #1:

Comment 1: The authors addressed all the issues highlighted in the previous report. The reviewer appreciate the experimental works done by the authors to explain the mechanistic aspects of the study. I suggest the manuscript for publication in Nature Communications.

Response: Thanks for all the constructive suggestions from this reviewer.

Reviewer #2:

Comment 1: Since the degradation of CD displayed radiation and radical dependent manner, it will be convincing to simply add radicals generated by small molecules into CD solutions for assessment of degradation in dark.

Response: Thanks for the suggestion. Following this reviewer's suggestion, we have now performed another CD degradation experiment in the dark. The Fenton reaction ($\text{H}_2\text{O}_2 + \text{FeCl}_2$) was applied to produce the hydroxyl radicals in the dark. There were totally seven treatments including the control treatment without any addition of H_2O_2 and FeCl_2 , 0.3 mM H_2O_2 , 0.3 mM H_2O_2 and 0.3 mM FeCl_2 , 1 mM H_2O_2 , 1 mM H_2O_2 and 1 mM FeCl_2 , 3 mM H_2O_2 , 3 mM H_2O_2 and 3 mM FeCl_2 . As shown in Supplementary Fig. S2a below, no hydroxyl radicals was generated in the treatments without any addition of FeCl_2 . When FeCl_2 was applied, a substantial amount of hydroxyl radicals was generated and their concentration increased with the increase in the concentration of H_2O_2 and FeCl_2 . In accordance with the production of hydroxyl radicals, more CDs degraded in the presence of FeCl_2 as compared to the respective control treatments containing the same concentration of H_2O_2 but without any addition of FeCl_2 , excluding the possibility that the degradation of the CDs in the dark was due to the oxidation of H_2O_2 itself. Further, more CDs degraded in the treatments with higher concentration of hydroxyl radicals. All these results clearly indicate that the hydroxyl radicals played important roles in the photodegradation of the CDs observed in the present study. Similar experiments were not performed for alkyl radicals as these radicals are complicated and have different species or structures, which were not identified herein. Without this detailed information about their speciation, we cannot simulate the effects of alkyl radicals on CD degradation in the dark. Detailed information about the methods and results of this degradation experiment in the dark has now been provided in Line 142-150 and Line 461-469.

Fig. S2 | Effects of hydroxyl radicals (·OH) on carbon dot (CD) degradation in the dark.

When 0.3–3 mM FeCl₂ was added to the experimental medium, a substantial amount of ·OH was generated, which increased with the increase in the concentration of H₂O₂ and FeCl₂. Following a similar trend as that of ·OH, more CDs degraded in the presence of FeCl₂ as compared to the respective control treatments with the same concentration of H₂O₂ but without any addition of FeCl₂. Further, more CDs degraded at higher concentrations of H₂O₂ and FeCl₂. The positive correlation between ·OH production and CD degradation indicates the important role of ·OH in CD degradation. Data are presented as the mean ± s.d. (n = 3 independent experiments). Source data are provided as a Source Data file.

Comment 2: A scheme should be added to facilitate the understanding of degradation mechanism.

A discussion of degradation mechanisms as well as the structure-activity relationships may strengthen the fundamental implication of this study.

Response: Thanks for the suggestion. Now a scheme has been provided in Supplementary Fig. S3 to facilitate the understanding about the degradation mechanism of the CDs. As shown in Supplementary Fig. S3 below, upon light irradiation, photogeneration of electrons and holes would occur on the surface of the CDs, resulting in the formation of hydroxyl ($\cdot\text{OH}$) and alkyl ($\cdot\text{R}$) radicals. Conjugated π -bond, surface defects, and functional groups of the CDs were also possibly involved in the generation of radicals. The radicals could then attack the CDs in multiple ways (*e.g.*, transferring electrons to CDs, abstracting H-atom from CDs, hydroxylating CDs by attacking the C=C bonds of CDs) and the CDs thus degraded into molecules containing hydroxyl (polyols), carbonyl, and benzene (benzenoids) groups. This information has now been clarified in Line 152-158. Nevertheless, the complete mechanism of CD photodegradation is still unknown and should be systematically examined in the future. Even the structures of CDs themselves are largely unknown. Therefore, we just proposed some potential degradation mechanisms of the CDs according to the data we obtained in the present study and based on the information about the photocatalytic activity of CDs in the literature.

Fig. S3 | Scheme of carbon dot (CD) photodegradation. Upon light irradiation, photogeneration of electrons and holes would occur on the surface of the CDs, resulting in the formation of hydroxyl ($\cdot\text{OH}$) and alkyl ($\cdot\text{R}$) radicals. Conjugated π -bond, surface defects, and functional groups of the CDs were also possibly involved in the generation of radicals. The radicals could then attack the CDs in multiple ways (*e.g.*, transferring electrons to CDs, abstracting H-atom from CDs, hydroxylating CDs by attacking the C=C bonds of CDs) and the

CDs thus degraded into molecules containing hydroxyl (polyols), carbonyl, and benzene (benzenoids) groups.

REVIEWERS' COMMENTS

Reviewer #2 (Remarks to the Author):

The revised manuscript has been significantly improved. I'm fully convinced by the new results.